# Endogenous oscillatory rhythms and interactive contingencies jointly influence infant attention during early infant-caregiver interaction

Emily AM Phillips[1]*, Louise Goupil[2], Megan Whitehorn[1], Emma Bruce-Gardyne[1], Florian A Csolsim[1], Navsheen Kaur[1], Emily Greenwood[1], Ira Marriott Haresign[1], Sam V Wass[1]

[1]Department of Psychology, University of East London, London, United Kingdom; [2]Centre National de la Recherche Scientifique, Laboratoire de Psychologie et NeuroCognition, Université Grenoble Alpes, Grenoble, France

## eLife assessment

This study reports **important** evidence that infants' internal factors guide children's attention, and that caregivers respond to infants' attentional shifts during caregiver-infant interactions. The authors analyzed EEG data and multiple types of behaviors using **solid** methodologies that can guide future studies of neural responses during social interaction in infants.

**Abstract** Almost all early cognitive development takes place in social contexts. At the moment, however, we know little about the neural and micro-interactive mechanisms that support infants' attention during social interactions. Recording EEG during naturalistic caregiver-infant interactions (N=66), we compare two different accounts. Traditional, didactic perspectives emphasise the role of the caregiver in structuring the interaction, whilst active learning models focus on motivational factors, endogenous to the infant, that guide their attention. Our results show that, already by 12 months, intrinsic cognitive processes control infants' attention: fluctuations in endogenous oscillatory neural activity associated with changes in infant attentiveness. In comparison, infant attention was not forwards-predicted by caregiver gaze or vocal behaviours. Instead, caregivers rapidly modulated their behaviours in response to changes in infant attention and cognitive engagement, and greater reactive changes associated with longer infant attention. Our findings suggest that shared attention develops through interactive but asymmetric, infant-led processes that operate across the caregiver-child dyad.

## Introduction

Almost all early cognitive development and learning takes place in social contexts (*Trevarthen, 2001*). We know that caregiver behaviours influence where, how, and for how long children allocate their attention in real-world settings (*Yu and Smith, 2016*), and that individual differences in how caregivers behave while interacting with their child can predict later language learning and socio-cognitive development (*Donnellan et al., 2020*; *Henning et al., 2005*; *Murray et al., 2016b*). But we currently understand little about the intrapersonal and bidirectional neural mechanisms that influence how infants allocate their attention to learn from their environment during naturalistic, free-flowing interactions.

**\*For correspondence:**
emily.phillips@bbk.ac.uk

**Competing interest:** The authors declare that no competing interests exist.

A number of different theoretical models try to explain how social partners influence infants' attention. The first, and probably the oldest, proposes that caregivers directly and didactically organise their infant's attention, by building a structure of how they pay attention, and when, and encouraging the child to follow their attentional focus (*Bornstein, 1985*). This might take place through children copying where caregivers are paying attention, second by second, while they complete a shared task (*Yu and Smith, 2016*). Or, it might happen through adults organisedly and actively using ostensive signalling to guide infant attention (*Csibra and Gergely, 2009*). To do this, the adult partner might be using salient behaviours (e.g. eye gaze, high pitched speech, etc.) to exogenously influence where children allocate their attention. In either of these cases, infant attention is *reactive* to changes in the behaviour of the caregiver (*Sebanz and Knoblich, 2009*; *Yu and Smith, 2013*).

Recent micro-behavioural analyses of caregiver and infant gaze behaviour during joint table-top interactions support this perspective, to some extent. Multimodal behavioural inputs by the caregiver are known to support episodes of sustained attention towards objects: for example, infant attention durations lasting over 3 s are directly predicted by the amount and timing of caregiver speech and touch to objects (*Suarez-Rivera et al., 2019*). More indirectly, other research has shown that infant attention is more fast-changing in joint compared to solo play, despite infant attention durations being, overall, longer in joint play (*Wass et al., 2018a*) - suggesting that endogenous cognitive processes such as attentional inertia the finding that, the longer a look lasts, the less likely it is to end *Richards and Anderson, 2004* have less of an influence on infant attention in social contexts. Further research suggests that, rather than following the focus of the adults' gaze, infants most often co-ordinate their attention with the adult through attending towards their partners' object manipulations, which corresponds to the idea that adults use exogenous attention capture to drive infant attention (*Yu and Smith, 2013*). Other salient behaviours might also be important, but are under-investigated. For example, infant-directed speech is known to contain more variability in amplitude and pitch (*Cooper and Aslin, 1994*), which increases its auditory salience (*Nencheva et al., 2019*), but although it is known that children generally pay more attention to infant-directed speech (*Nencheva et al., 2019*), no previous research has examined whether caregivers use moment-by-moment variability in the salience of their voice to influence how children allocate attention.

Within this framework, it is possible that, rather than repeated and reactive contingent responsivity to isolated behaviours, temporal dependency between infant and caregiver attention is driven by infant behaviour becoming periodically coupled to the behavioural modulations of their partner (*Meyer et al., 2019*; *Wass et al., 2022*). Similar to inter-dyadic patterns of vocalisations in adults and marmoset monkeys (*Takahashi et al., 2013*; *Wilson and Wilson, 2005*), in early infant-caregiver interactions, vocal pauses in one partner's vocalisations can be predicted from those of the other (*Gratier et al., 2015*; *Jaffe et al., 2001*), and, during face-to-face interactions at the end of the first year, caregiver-infant facial affect becomes temporally aligned (*Feldman et al., 1999*). Oscillatory entrainment, that is, consistent temporal alignment between fluctuations in caregiver and infant behaviour, could be particularly important in ensuring that salient sensory and information-rich inputs by the caregiver occur at moments infant are most receptive to receiving information (*Wass et al., 2022*).

An alternative interpretation of these micro-behavioural findings, however, is that, rather than structuring infant behaviour through leading infant attention, caregivers instead influence how infants pay attention by following and responding to re-orientations in their infant's attention. This second model suggests that, rather than considering unidirectional caregiver->child influences we should instead be considering bidirectional child<->caregiver influences. In following the focus of their infants' attention at moments that they reorient towards a new object, the caregiver 'catches' and extends infant attention with reactive and dynamic change in their salient ostensive behaviours, to which infants are responsive (*Yu and Smith, 2016*). The contingent adaptation of the caregiver to modulations in infant attention serves to maintain and extend infant attention, and provides inputs at points where infants anticipate to receive new information (*Yu and Smith, 2012*). Indeed, from early infancy, caregivers are contingently responsive to modulations in their infant's behaviour. From 2 to 3 months, caregivers respond differentially to distinct facial affects produced by the infant (*Murray et al., 2016a*), modulate their vocal feedback to infant babbling (*Albert et al., 2018*; *Goldstein and Schwade, 2008*; *Yoo et al., 2018*); and, towards the end of the first year, provide more labelling responses relative to infant's pointing than to their object-directed vocalisations (*Wu and Gros-Louis, 2015*).

According to the first model, then, caregivers drive and actively control infants' attention during joint interaction. According to the alternative model, caregivers influence infants' attention by reactively and contingently responding to the infant's attention shifts. But according to the latter model, what drives how infants initially allocate their attention in the first place? Recording infant EEG activity during naturalistic caregiver-infant interactions is one way that we can begin to understand this, by providing a method to examine endogenous fluctuations in top-down attention control processes which might be otherwise unobservable behaviourally.

In adults, the timing of attention shifts can be partially described using an oscillatory structure, reflecting rhythmic attention reorientations that possibly correspond to fluctuations in the CNS (*Nuthmann et al., 2010b*; *Nuthmann and Henderson, 2010a*; *Nuthmann and Matthias, 2014*). Research with infants has also suggested that, even during early life, infants' attention shifting is not purely stochastic (*Robertson, 2014*). In free-viewing paradigms, infant gaze exhibits a fractal structure that increases over the course of the first year (*Stallworthy et al., 2020*), and, periodic structure in 12-month-old attention patterns has been associated with increased cognitive control (*Feldman and Mayes, 1999*). Regulatory mechanisms endogenous to the infant could therefore be one mechanism that influences when infants reorient their attention during real-world naturalistic interactions.

By the end of the first year, however, as well as periodic attention reorientations, fluctuations in top-down attentional control processes, thought to be driven by the executive attention system, begin to influence where and when infants shift their attention. Recent research has shown that infants routinely deploy active and effortful information-sampling strategies to maximise their opportunities for learning (*Gottlieb et al., 2013*; *Goupil and Proust, 2023*; *Kidd et al., 2012*; *Oudeyer and Smith, 2016*; *Poli et al., 2020*). For example, infants aged 8–9 months optimise information gain by directing their attention towards stimuli that are neither too complex, nor too predictable (*Kidd et al., 2012*; *Kidd et al., 2014*) and disengage from stimuli that are less informative compared to past observations (*Poli et al., 2020*). Corresponding to developments in intentionally mediated forms of joint communication (*Tomasello et al., 2007*), infants are also thought to begin to use active strategies to directly elicit information from a social partner about their environment. For example, infants aged 12–14 months point in an interrogative manner (*Begus and Southgate, 2012*; *Kovács et al., 2014*), and look towards their caregiver to ask for help when uncertain (*Bazhydai et al., 2020*; *Goupil et al., 2016*).

These approaches suggest that infants' endogenous engagement or interest forward-predicts their attention patterns. In addition, though, there is an alternative, complementary possibility. Infants' attention shifts may initially happen as random, regulatory behaviours (*Robertson, 2014*; *Stephens and Charnov, 1982*; i.e. not forward-predicted by fluctuations in infants' endogenous engagement or interest); processes *after* the attention shift (determined by what information is present at the attended-to location) may drive increases in infants' endogenous engagement or interest which prolong that attention episode. (This distinction is similar to that we discussed above, about whether caregiver behaviours forwards-predict infant attention, or whether caregivers influence infants by reactively responding to their attention shifts, but operates at the individual level.) Consistent with this possibility, dynamic, generative models based on this framework can accurately predict attention patterns at least in younger infants (*Robertson, 2014*; *Robertson et al., 2004*).

To examine how fluctuations in endogenous engagement drive and/or maintain infant attention during naturalistic interactions, we can record EEG. EEG provides a non-invasive method for measuring oscillatory neural activity, which is thought to reflect patterns of neural firing and communication in the brain (*Cole and Voytek, 2017*). Oscillatory activity is grouped into different frequency bands, corresponding to delta (1–3 Hz), theta (3–6 Hz), alpha (6–9 Hz), beta (9–20 Hz), and gamma (>20 Hz) in infancy (*Cohen, 2014*). Given that top-down influences on attention are difficult to measure behaviourally, from, for example, the timing of an attention shift, examining infants' neural activity during social interactions provides a way to measure intrinsically guided processes, that influence where and when infants allocate their attention during online interactions.

In particular, EEG activity in the theta range has been found to increase before and during episodes of endogenously controlled attention in early infancy. For example, in the first year of infancy, theta activity increases during self-guided object exploration (*Orekhova et al., 2006*), associates with the length of sustained episodes of attention to individual objects (*Wass et al., 2018b*), and predicts infants' performance in tasks requiring top-down attention control, such as attending towards

dynamic, screen-based stimuli and anticipating the next actions of an experimenter (*Orekhova et al., 2006*; *Xie et al., 2018*). Corresponding to the associations between endogenous attention control and theta activity observed in adults, these effects are most often localised over fronto-central areas (*Cohen et al., 2012*; *Jones et al., 2020*). Measuring fluctuations in infant EEG theta oscillations during naturalistic social interactions is therefore particularly informative to examining how top-down attention control processes associate with infant gaze behaviours during early infant-caregiver interactions.

Corresponding to the active learning accounts outlined above, one possibility is that endogenous engagement or interest largely drives the allocation of infant attention during shared interactions. If so, we would predict that increases in theta activity forward-predict increases in infant attention (i.e. increases in theta activity precede increases in infant attentiveness). Indeed, in non-interactive settings, theta activity over fronto-central electrodes has been found to increase during episodes of anticipatory attention, and theta oscillations occurring in the time before infants look towards an object predicts the length of time infants pay attention to that object during solitary play (*Orekhova et al., 2006*; *Wass et al., 2018b*). Considering the alternative possibility – that fluctuations in endogenous cognitive processing that occur in the time after an attention shift are important to sustaining infant attention during online social interactions – we would predict fluctuations in theta activity after an attention shift to associate with the length of infant attention durations. Related to this possibility, Xie and colleagues found that, whilst 10- to 12-month-old infants attended towards cartoon videos, theta activity increased during heart-rate defined periods of attentional engagement (*Xie et al., 2018*; see also *Jones et al., 2020*).

In summary, research has examined two influences that could support how infants pay attention in social settings. The first type of influence is endogenous engagement or interest. The second is caregivers' behaviour, which is exogenous to the infant. But for both of them, it is unclear whether the influences are largely forwards-predictive or reactive. Does infants' endogenous attention engagement forwards-predict attention, or do fluctuations in engagement that take place after an attention shift predict how long that episode lasts? And do caregivers drive infant attention using salience cues, or do they reactively change their behaviours in response to infant behaviours? Examining these questions using neurocognitive techniques is crucial to understanding the endogenous and interactive mechanisms that drive infant attention and learning in shared, naturalistic interactions.

Here, recording EEG from infants during naturalistic interactions with their caregiver, we examined the (inter)-dependent influences of infants' endogenous oscillatory neural activity, and inter-dyadic behavioural contingencies in organising infant attention. First, we examined processes endogenous to the infant that determine the timing of their attention during the interaction (part 2). Second, we examine fluctuations in caregiver behaviours (part 3).

In part 1, we first test whether oscillatory structures can be derived from the patterns of infant and caregiver looking behaviour at an individual level, by computing the partial auto-correlation function (PACF) for caregiver and infant attention onsets. The PACF measures the excess correlation of a time-series with itself at different lag-intervals in time, controlling for the correlations of all other shorter time lags (*Chatfield, 2004*). If infant looking behaviour is characterised by an oscillatory structure, then we would predict a peak in the PACF, reflecting that look durations are largely consistent in length. We then test whether infant and caregiver behaviours act as coupled oscillators, by examining the time-course of the cross-correlation function between the timing of infant and adult attention onsets (*Takahashi et al., 2013*). A cross-correlation function measures the association of one time series with another time series at different lag-intervals in time (*Chatfield, 2004*). If caregiver and infant attention durations act as coupled oscillators, we would expect regular peaks in the cross-correlation function, reflecting the alignment of infant and caregiver attention onsets at consistent time intervals. This would point to the existence of mechanisms of influence between infant and adult gaze that our other analyses, examining forwards- and backwards-predictive associations (see below), would be unable to detect.

In part 2 we then assess whether infants' endogenous cognitive processing forward-predicts infant attention, by using cross-correlations to estimate the forwards- and backwards-predictive associations between infant theta activity, recorded over fronto-central electrodes (*Braithwaite et al., 2020*), and infants' attention durations. To produce a continuous index of infant attention durations, necessary for conducting the cross-correlation analysis, we include all infant looks to objects and their partner in this section. Whilst attention to objects vs. faces has been found to elicit differential patterns of neural

activity (*Conte et al., 2020*; *Hoehl et al., 2008*), here, corresponding to the association of sustained attention episodes to increased engagement in early infancy (*Richards, 2010*), we assess whether longer attention durations towards objects and partners associate with overall increases in infants' endogenous cognitive processing. If infant theta activity forward-predicts increases in infant look durations, we would predict an asymmetry in the shape of the cross-correlation function, comparing positive vs. negative lags, with increased associations at negative lags (*de Barbaro et al., 2017*; *Wass et al., 2018a*). In addition to this, we further examined reactive changes in infant endogenous oscillatory neural activity that take place after the onset of infant attention episodes towards objects only. To do so, we used three analyses: first, using linear-mixed effects models, we examined the direct temporal associations between infant attention durations and the average levels of infant theta activity during that look. If endogenous cognitive processes after the onset of infant attention are important in maintaining their attention towards objects, then we would expect longer looks to associate with greater average theta activity. Second, to assess whether reactive changes in infant theta activity occurred immediately after look onsets towards objects, we time-locked infant theta-activity to the onset of each object attention episode and examined the continuous changes in infant theta activity in the 5 s after look onset. Finally, to test whether endogenous cognitive processing increases over the course of an attention episode, we examined how theta activity changes dynamically over the course of individual looks. This was done by averaging infant theta activity in three sperate chunks over the duration of each attention episode (i.e. the first, middle and last part of the object look), and comparing between the averaged chunks.

In part 3 we examine the (inter)-dependent associations between caregiver behaviours and infant attention. We examine two aspects of caregiver behaviour in particular. In the first section of part 3, we examine caregiver gaze behaviour, using the same cross-correlation analysis conducted in part 2, to test whether increases in caregiver attention durations forwards- or backwards-predicted changes in infant attention durations. Again, if increases in caregiver attention durations forward-predicts increases in infant attention durations, we would predict an asymmetry in the cross-correlation function, with increased associations between infant attention and caregiver behaviour at negative time-lags (*de Barbaro et al., 2017*; *Wass et al., 2018a*). In order to test whether any association between infant attention and caregiver behaviour was independent of the association between infant attention and their endogenous oscillatory neural activity, we also conducted cross-correlations to examine the associations between caregiver attention durations and infant theta activity. And we used the same two analyses as used in part 2 to examine how caregiver gaze behaviour changes reactively following the onset of an infant attention episode towards objects. First, we use linear mixed effects models to examine whether longer infant look durations associate with longer look durations by the caregiver, second, we assess immediate change in caregiver gaze behaviours in the 5 s after infant object look onsets, and, finally, we examine whether caregiver attention durations change significantly from the first to the last third of an object-directed attention episode.

In the second section of part 3, we examined saliency in the caregiver's speech signal by computing the rate of change in the fundamental frequency of their voice (*Nencheva et al., 2019*). For this, we used the same analysis approach. First, we conducted cross-correlations to examine whether changes in caregiver vocal saliency forwards- or backwards-predict changes in infant attention durations. Second, we examined how caregiver vocal saliency changes reactively following the onset of an infant attention episode. Allostatic attentional-structuring models predict reactive change in caregiver behaviour at the onset of infant attention, and over the duration of the look, that associate with the length of infant looking.

## Results

The results section is divided into three parts. In part 1, we first conduct descriptive statistics of infant attention durations, and test for oscillatory structures in caregiver and infant attention. Then, in part 2, we examine whether endogenous infant neural activity forwards-predicts fluctuations in infant attention, and/or reactively changes in the time after the onset of an object-directed attention episode. In part 3, we assess whether modulations in caregiver gaze and vocal behaviour forwards-predict fluctuations in infant attention, and/or reactively change in the time after infants shift their attention towards a new object.

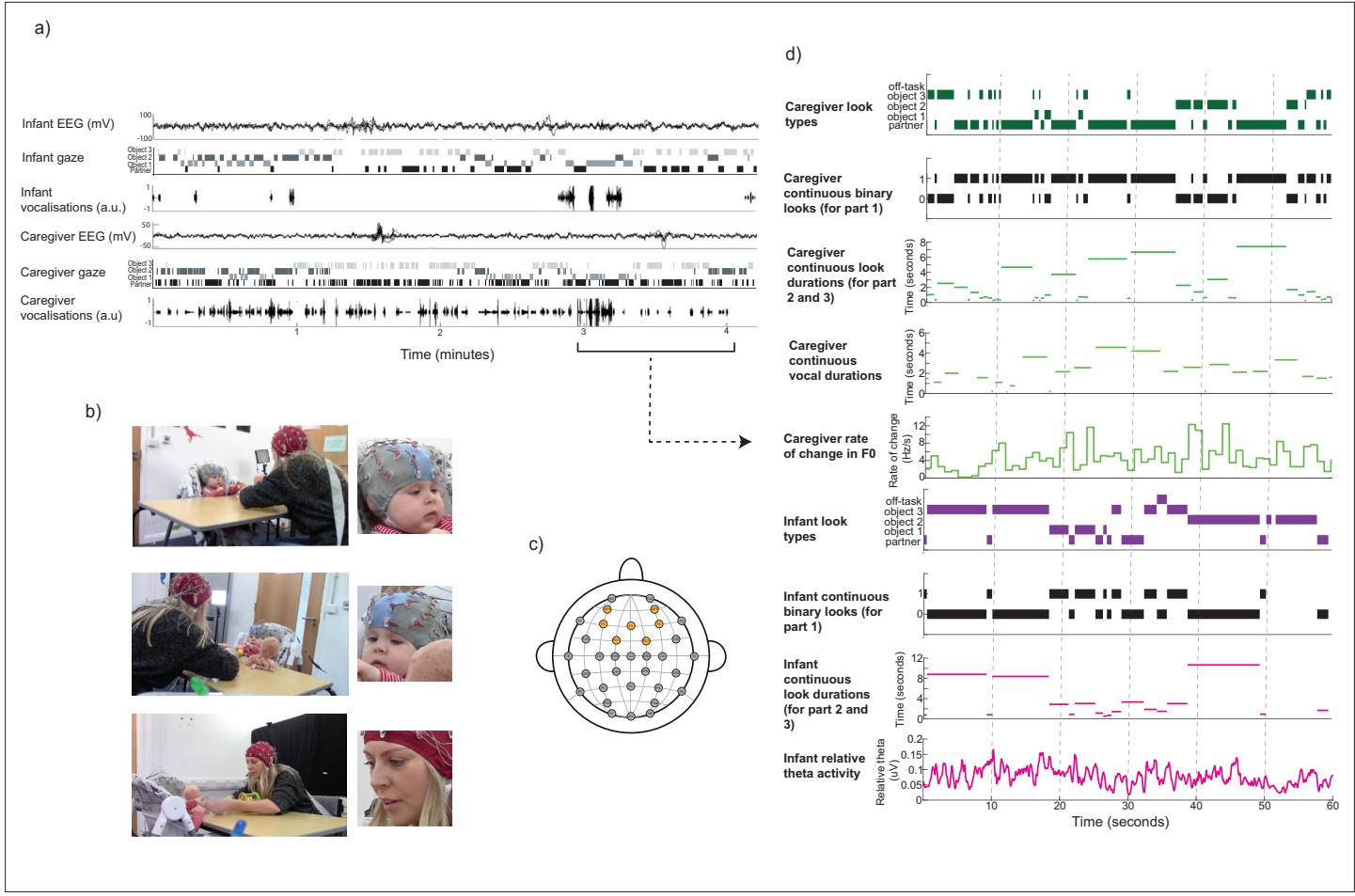

**Figure 1.** Experimental set-up and example of continuous variables. (**a**) Raw data sample, showing (from top) infant EEG over fronto-central electrodes, after pre-processing, infant gaze behaviour, infant vocalisations, caregiver EEG over fronto-central electrodes, caregiver gaze behaviour, caregiver vocalisations. (**b**) Example camera angles for caregiver and infant (right and left), as well as zoomed-in images of caregiver and infant faces, used for coding. (**c**) Topographical map showing electrode locations on the bio-semi 32-cap; fronto-central electrodes included in the theta activity analysis are highlighted in orange (AF3, AF4, FC1, FC2, F3, F4, Fz). (**d**) Continuous behaviour and EEG variables extracted from the caregiver and infant time-series, showing (from top), caregiver looks to objects, the partner, and off-task looks, caregiver binary attention durations (for part 1), caregiver continuous attention durations (for part 2 and 3), caregiver vocalisation durations, rate of change in caregiver F0, infant looks to objects, their partner, and off-task looks, infant binary attention durations (for part 1), infant continuous look durations (for part 2 and 3), infant relative theta activity.

## Part 1 - Oscillatory structures in caregiver and infant attention

First, as descriptive statistics, we report on the frequency distribution of caregiver and infant attention durations towards objects, the partner, and periods of off-task attention, dividing attention durations into 100ms bins. Histograms showing the distribution of caregiver and infant attention durations towards objects, partners, and non-targets are displayed in *Figure 1*; *Figure 2a*. In both distributions the mode is greater than the minimum value, consistent with previous observations that attention shifting is periodic (*Saez de Urabain et al., 2017*). The caregivers' distribution is also more left-skewed compared to the infants' distribution, reflecting the shorter and more frequent attention durations by the caregiver c.f (*Wass et al., 2018b*; *Yu and Smith, 2012*). Finally, consistent with previous reports (*Yu and Smith, 2013*), caregivers tended to look towards their partner more frequently than infants, with infants attending most frequently to the objects (*Figure 2a*).

Next, to investigate whether there was an oscillatory component in the caregiver and infant gaze time series, we computed the PACF of a binary attention variable separately for caregiver and infant using 100, 200, 500, and 1000ms lags (see Materials and methods and *Figure 1d* for more detail). In order to explore whether the PACF reflected the temporal interdependencies between infant/caregiver attention episodes (i.e. how likely an attention episode of a given length was to be followed

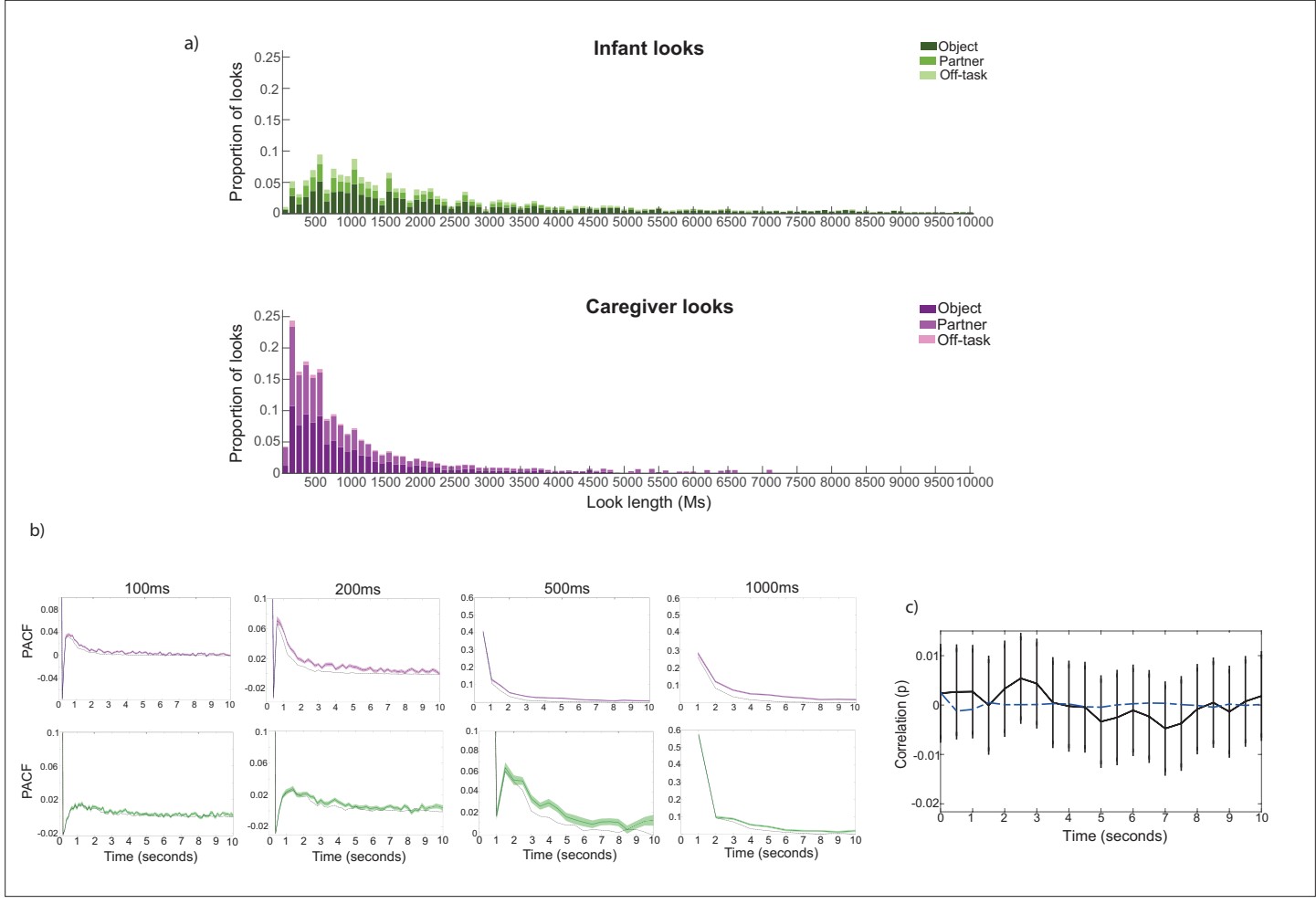

**Figure 2.** Testing for oscillatory patterns of attention behaviour in infants and caregivers. (**a**) Histogram of caregiver and infant attention episodes to objects, their partner and off-task episodes. Stacked bars show the number of episodes in each category for each 100ms bin for all episodes up to 10 s in duration. (**b**) PACF computed at different time lags for caregiver and infant gaze time series. Coloured lines show the PACF for infants (green) and caregivers (pink); shaded areas show the SEM. Dashed black lines show the PACF of shuffled attention duration data. (**c**) Cross-correlation between caregiver and infant binary gaze variables. Black line shows the Spearman correlation coefficient at time-lags ranging from 0 to 10 s; error bars indicate the SEM. Blue dashed line shows the permutation cross-correlation between two time series of poisson point process; one matching the average look rate of caregivers and the other, the average look rate of infants. N=66 for all analyses.

by another of a similar length), or, more simply, the overall distributions of attention episodes (i.e. how common attention episodes of a given length are overall), the PACF was repeated after shuffling the infant and caregiver attention durations in time (see Materials and methods for permutation procedure).

*Figure 2b* shows that for the 100ms, 200ms time bins (infant and caregiver) and 500ms time bin (infant only), a second peak in the PACF occurs after the lag 1 term. This pattern is also observed in the baseline data (in which looks have been randomly shuffled in time). It reflects therefore, the overall pattern already shown in the histograms in (*Figure 2a*), that infants and adults shift their attention at regular time intervals, and that caregiver looks are generally shorter compared to infant looks. It can also be seen that, at higher time lags, the observed PACF values are above the baseline rate. This indicates temporal interdependencies between look durations (i.e. that an attention episode of a given length is likely to be followed by another of a similar length), which are not present when the look durations are randomly shuffled to generate the baseline data.

Finally, we replicated previous analyses (*Takahashi et al., 2013*) to explore whether the inter-dyadic dynamics of caregiver and infant looking behaviour could be modelled as coupled oscillators. To do so, we first computed a binary attention variable, separately, for the infant and the caregiver. For this

binary attention variable, we coded each look alternatively as a 0 or 1 from the first look of the interaction to the last, irrespective of where the infant or caregiver was looking (e.g. objects/ the partner; see *Figure 1d*). To examine whether caregiver and infant attention changes at consistent temporal latencies, as would be the case if they were acting as entrained oscillators (*Wass et al., 2022*), we calculated the cross-correlation function between the infant and caregiver binary attention variables. If caregiver and infant gaze behaviour act as coupled oscillators, then the cross-correlation function should display significant peaks at regular intervals, reflecting these consistent latencies between attention shifts (*Takahashi et al., 2013*). In order to identify where peaks in the cross-correlation function exceeded chance, we computed Poisson point process timeseries with look duration lengths matching the average look duration in the actual data (see Materials and methods for more details). *Figure 2c* shows the results of this. Cluster-based permutation analysis revealed no significant peaks in the cross-correlation function, compared to baselines created through poisson process.

In summary, oscillatory mechanisms appear to govern both caregiver and infant attention durations; with infant attention durations centring around 1–2 s in length, and adults around 200–500ms. The cross-correlation analysis, however, suggested that caregiver and infant attention shifts do not act as coupled oscillators across the dyad (*Figure 2c*).

## Part 2 - Does endogenous neural activity forwards-predict infant attention, or reactively change following the onset of a new infant attention episode?

In this section, we investigate the association between infant endogenous oscillatory neural activity and infant attention durations, considering both forwards-predictive associations and reactive changes

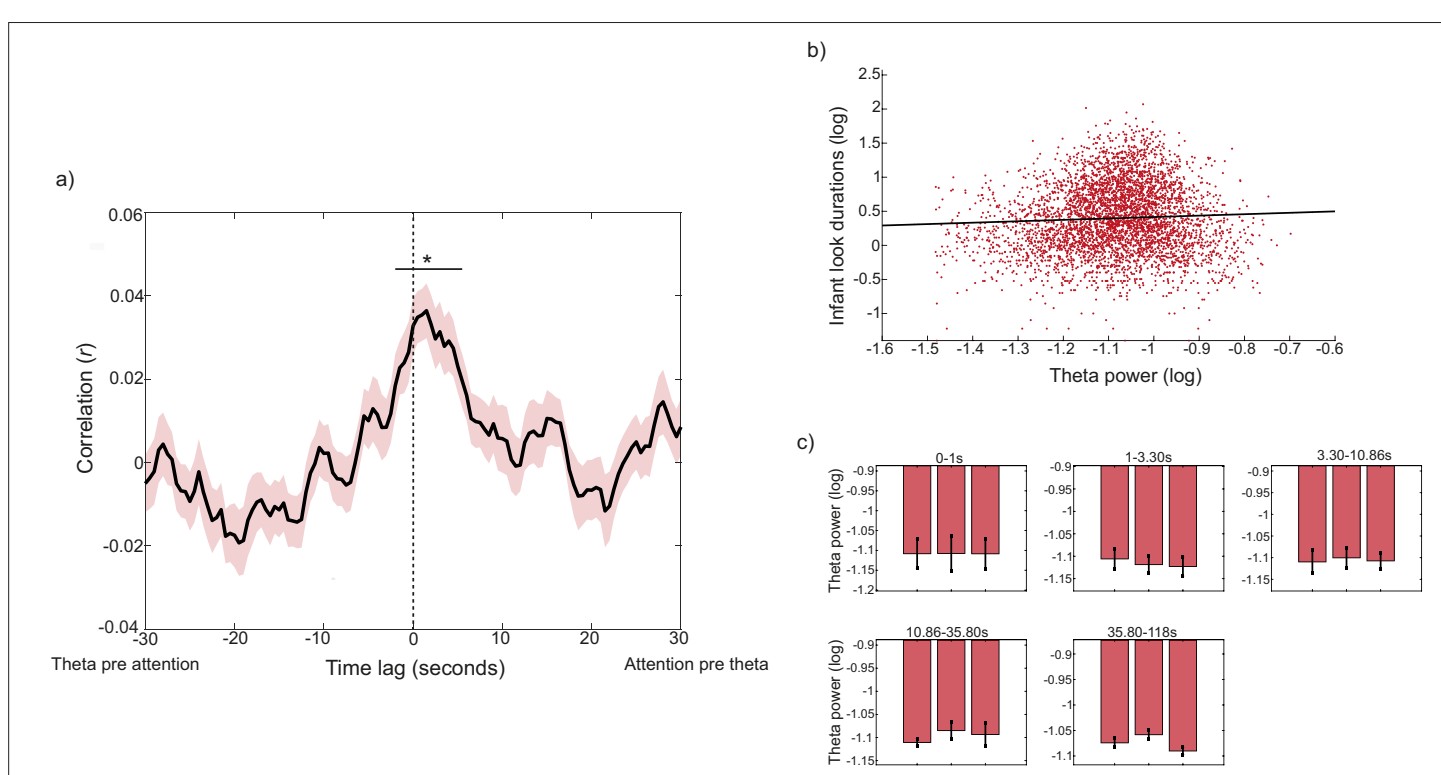

**Figure 3.** Association between infant attention durations and infant theta activity. (**a**) Cross-correlation between infant theta activity and infant attention durations. Black line shows the Pearson correlation at each time lag, coloured shaded areas indicate the SEM. Significant time lags identified by the cluster-based permutation analysis are indicated by black horizontal lines (*p<0.05). Cluster-based permutation analysis revealed a significant cluster of time points ranging from –2 to +6 seconds (p=0.004). (**b**) Linear mixed effects model, predicting infant object attention durations from infant theta activity. The model reveals a significant positive association between infant theta activity and attention durations ($\beta$=0.33; p<0.001, Spearman's *r*=0.04, p=0.004; Pearson's *r*=0.05, p<0.001). (**c**) Infant theta activity split into 3 attention chunks across the duration of attention episodes, binned according to episode length. Wilcoxon signed ranks tests explored significant differences between attention chunks, for each duration bin (*p<0.05). N=60 for all analyses.

in infants' endogenous oscillatory activity after the onset of object-directed attention episodes. See Materials and methods section *Infant EEG artifact rejection and pre-processing* for details of our EEG pre-processing pipeline, including a description of the specially designed and automatic movement artifact rejection procedures.

### Forwards predictive association between infant attention and infant theta activity

To examine whether infant endogenous neural activity significantly forwards-predicted infant attentiveness, we calculated a cross-correlation between the continuous infant attention duration time-series (see *Figure 1d*), including all infant attention episodes to objects, the partner and looks elsewhere, and infant theta activity. *Figure 3a* shows the results of the cross-correlation analysis. This analysis revealed a significant, positive association between the two variables at time-lags ranging from –2 to +6 s (p=0.004).

This indicates that infant theta power significantly forwards-predicted infant attention durations at lags up to 2 s, as well as that infant attention durations significantly forwards-predicted infant theta at lags of up to 6 s. Interpreting the exact time intervals over which a cross-correlation is significant is challenging due to the auto-correlation in the data (*Clifford et al., 1989*; *Pickup, 2014*), but there are two points of significance here. The first is the fact that the peak cross-correlation is observed not at time 0 but at time t+1.5 s (i.e. between looking behaviour at time t and theta power at time t+1.5 s). The second is that the significance window is asymmetric around time 0. Neither of these points can be attributable to residual auto-correlation. Overall, then, we can conclude that there is a temporally specific association between infant attention durations and theta power; and that attention durations forwards-predict theta power more than vice versa.

### Reactive change in infant theta activity following look onset

In addition to the continuous cross-correlation analysis, including all infant looks, we conducted two further analyses to investigate the association between theta activity and the duration of infant attention episodes towards objects. First, we calculated a linear mixed effects model to examine the association between the lengths of infant object attention episodes and average theta activity during that episode. The model predicted infant attention durations from infant theta activity, as a fixed effect, including participant as a random factor (see Materials and methods for more detail). This showed a significant, positive association between the two variables ($\beta$=0.33; p<0.001); scatter plot between the two variables is shown in *Figure 3b*. This indicates that higher average theta power across the attention episode associates with longer attention durations. Second, we explored dynamic change in theta activity relative to the onset of infant attention episodes towards objects. The modulation analysis (*Figure 3c*), examining average infant theta activity during each third of a continuous look, showed that there was little change in infant theta activity over the duration of infant attention episodes, for any duration time-bin: a series of Wilcoxon signed rank tests indicated decreases in infant theta activity for attention episodes lasting 1–3 s, but this did not survive Benjamini-Hochberg correction.

### Summary

In summary, there is a temporally specific association between infant attention durations and theta power, with attention durations forwards-predicting theta power more than vice versa (*Figure 3a*). Longer object attention episodes are associated with increased average theta activity over the length of the episode (*Figure 3b*), but little dynamical change in theta activity is observed over the course of an object-directed attention episode.

## Part 3 - Does caregiver behaviour forwards-predict infant attention, or reactively change following the onset of a new infant attention episode?

First, we examine whether caregiver gaze behaviour associates with infant attentiveness. Second, we examine whether caregiver vocal behaviour associates with infant attentiveness, focusing on the rate of change of caregiver F0 as an index of auditory salience. In each case, we examine both

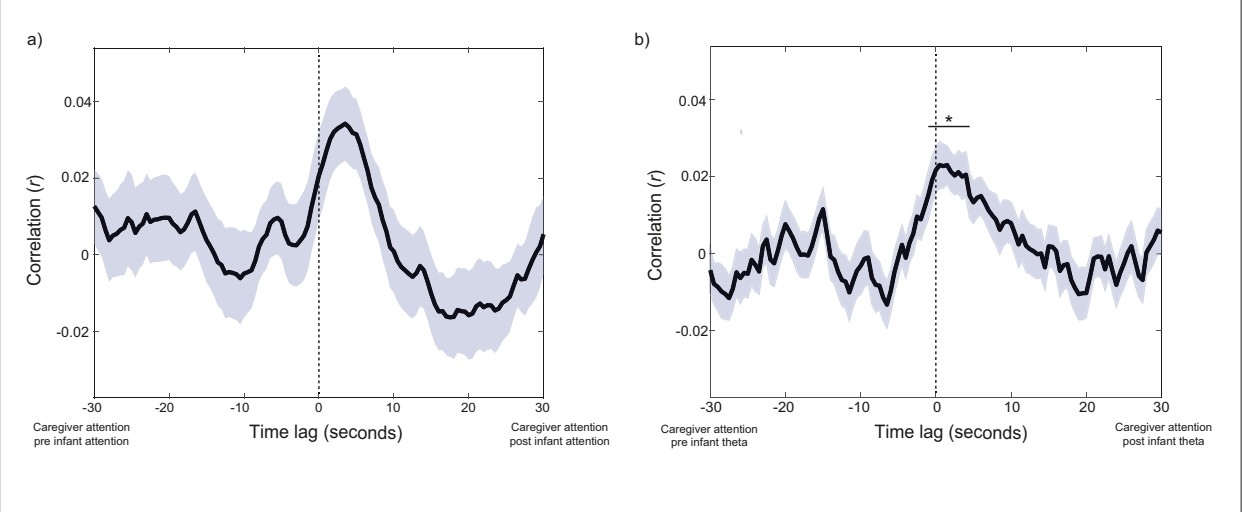

**Figure 4.** Assessing forwards-predictive associations between caregiver attention durations, infant attention durations, and infant theta activity. Black lines show the Pearson cross-correlation between two variables; coloured shaded areas indicate the SEM. Black horizontal lines show significant clusters of time lags (*p<0.05). (**a**) Infant and caregiver continuous attention durations (N=66). Cluster based permutation analysis revealed no significant clusters of time points, although one cluster verged on significance (p=0.10). (**b**) Infant theta activity and caregiver continuous attention durations (N=60). Cluster-based permutation analysis indicated one significant cluster ranging from –1–5 s (p=0.012).

forwards-predictive associations between caregiver behaviour and infant attention, and reactive changes in caregiver behaviour relative to the onsets of infant attention episodes towards objects.

## Caregiver gaze behaviour

### Forwards-predictive associations between infant attention durations and caregiver attention durations

To examine whether caregiver attention forwards-predicts infant attentiveness, we conducted cross-correlation analyses between the continuous infant and caregiver attention durations (see *Figure 1d*). In order to test whether any association between infant attentiveness and caregiver attentiveness was independent of the association between infant attentiveness and their endogenous oscillatory neural activity shown in *Figure 3a*, we also repeated these analyses relative to infant theta activity. Results are reported in *Figure 4*. The cross-correlation between caregiver and infant attention durations peaks after lag zero (t+2.5 s), but cluster-based permutation analysis revealed no significant clusters of time points, although one cluster verged on significance (p=0.10). The cross-correlation function between caregiver attention durations and infant theta activity revealed a similar pattern (*Figure 4b*), peaking in the period after time 0, and the cluster-based permutation analysis revealed a significant cluster ranging from –1–5 s (p=0.012). Although it is likely that the association between caregiver attention durations and infant theta shown in *Figure 4b* is mediated by the association between caregiver attention durations and infant attention durations shown in *Figure 4a*, the latter association is significant whereas the former is not. As for the analyses described in part 2, the exact time window over which the cross-correlation is significant cannot be interpreted due to autocorrelation in the data; but the fact that the peak cross-correlation is observed, again, at t+1.5 s, and that the significance window is asymmetric around time 0, both indicate that, overall, infant theta predicts caregiver attention durations more than vice versa.

### Reactive change in caregiver look durations following infant look onset

To examine reactive change in caregiver attention to objects following the onsets of infant attention episodes to objects, we time-locked caregiver attention durations to infant attention onsets towards objects. *Figure 5a* shows changes in caregiver attention durations around the onset of infant attention towards an object. Cluster-based permutation analysis revealed a significant cluster of time points 0–4 s post attention onset (p=0.009), indicating that caregiver attention durations significantly decreased after the onset of a new infant attention episode. *Figure 5b* shows the same event-related

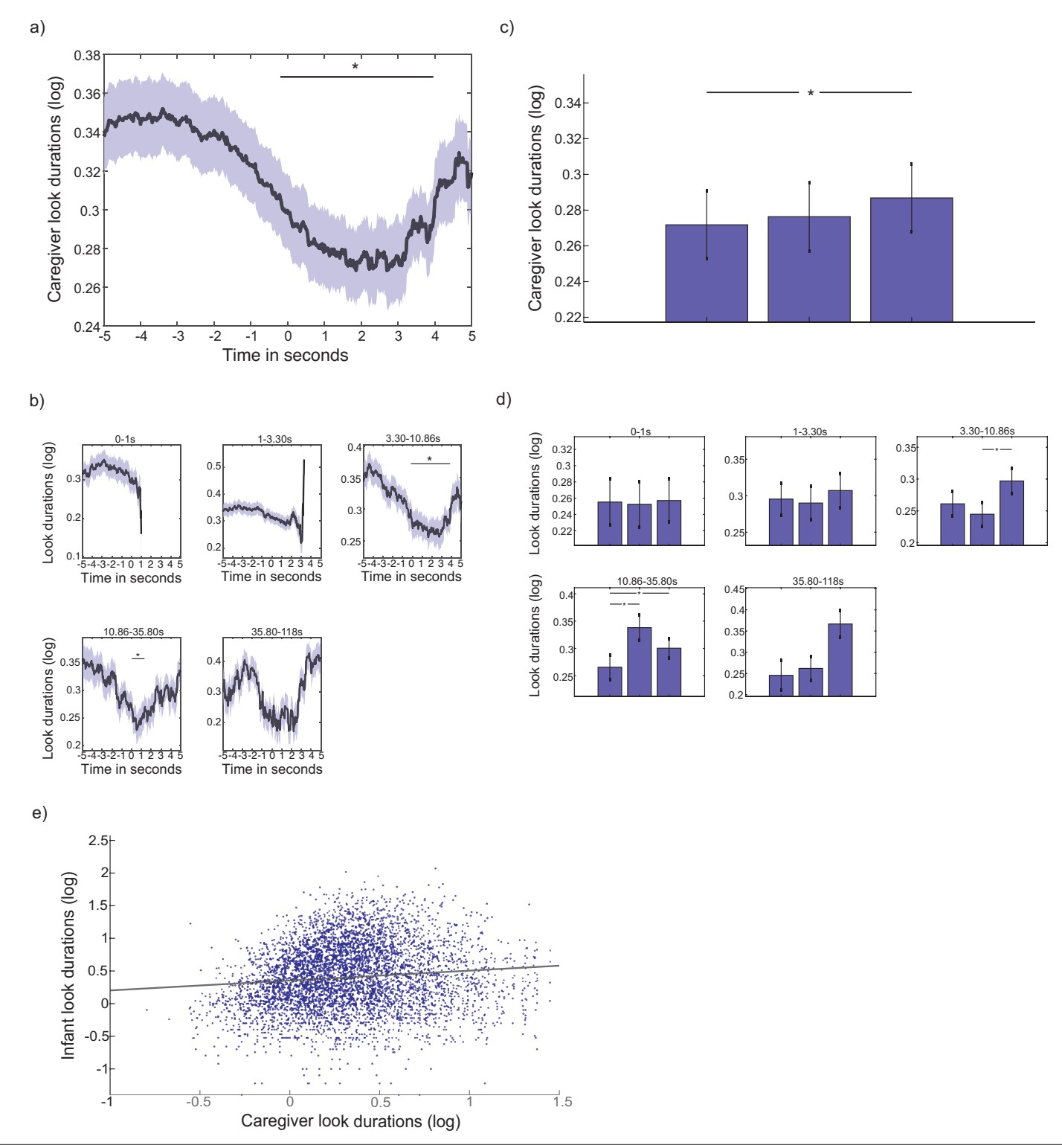

**Figure 5.** Dynamic event-locked association between infant object attention and caregiver attention durations. (**a**) Event-related analysis showing change in caregiver attention durations around infant attention onsets to objects: black line shows average caregiver attention durations (log); coloured shaded areas indicate the SEM. Black horizontal line shows areas of significance revealed by the cluster-based permutation analysis (*p<0.05). Cluster-based permutation analysis reveals a significant cluster of time points 0–4 s after attention onset (p=0.009). (**b**) event-related analysis split by infant object attention-duration time bins. Black lines show average caregiver attention durations (log); coloured shaded areas indicate the SEM. Black horizontal lines shows areas of significance revealed by the cluster-based permutation analysis (*p<0.05). Permutation analysis again revealed a

*Figure 5 continued on next page*

*Figure 5 continued*

decrease in caregiver attention durations in the time after attention onset for looks 3–35 s long. (**c**) Modulation analysis: each bar shows the median caregiver attention duration, across participants, for each chunk, averaged across all infant object attention durations. Wilcoxon signed ranks tests investigated significant differences between chunks (*p<0.05). (**d**) same as (**c**), for each infant object attention duration time bin. (**e**) Scatter-plot showing the association between infant object attention durations and caregiver continuous attention durations. Coloured dots show each individual object attention duration; black line shows the linear line of best fit. Linear mixed effects modelling revealed a significant positive association between the two variables (β=0.16, p<0.001 [Spearman's *r*=0.12, p<0.001, Pearson's *r*=0.11, p<0.001]). N=66 for all analyses.

analysis subdivided by infant attention duration. This revealed that the decrease in caregiver attention durations after infant attention onsets was significant for attention episodes lasting over 3 s.

To investigate how caregiver behaviour changed over the course of infant object looks, we next employed the same modulation analysis as described in part 2, computing differences in mean caregiver attention durations between 3 equal-spaced chunks over the course of an infant object look. This analysis revealed that, in contrast to the first 4 s of an infant attention episode during which caregiver attention durations decreased, caregiver attention durations actually significantly increased over the course of the entire attention episode, with a Wilcoxon signed ranks test indicating a significant difference between the first chunk of an attention episode and the third (*Figure 5c*). Dividing infant attention durations into log-spaced bins again revealed that this effect was driven by attention episodes lasting over 3 s (*Figure 5d*). Finally, we computed a linear mixed effect model to examine the association between infant object attention durations and caregiver object attention durations. The model predicted infant attention durations from caregiver attention durations, as a fixed effect, including participant as a random factor. Corresponding to the modulation analyses reported above, when we averaged over the course of the entire infant object attention episode, we found that longer infant object attention durations associated with longer average caregiver attention durations (*β*=0.16, p<0.001). *Figure 5e* shows the scatter plot of the association between infant look durations and averaged caregiver look durations over the length of each individual infant look duration.

## Summary

In summary, both the continuous and event-related analyses revealed that caregivers dynamically adapted their gaze behaviour in response to changes in infant attentiveness during the interaction. Infant theta activity significantly forwards-predicted caregiver attention durations, suggesting that caregivers dynamically adapt their behaviour according to infant engagement (*Figure 4b*). Caregiver attention durations to objects decreased around the start of a new infant attention episode (*Figure 5a*), but overall, longer infant attention durations associated with longer attention durations by

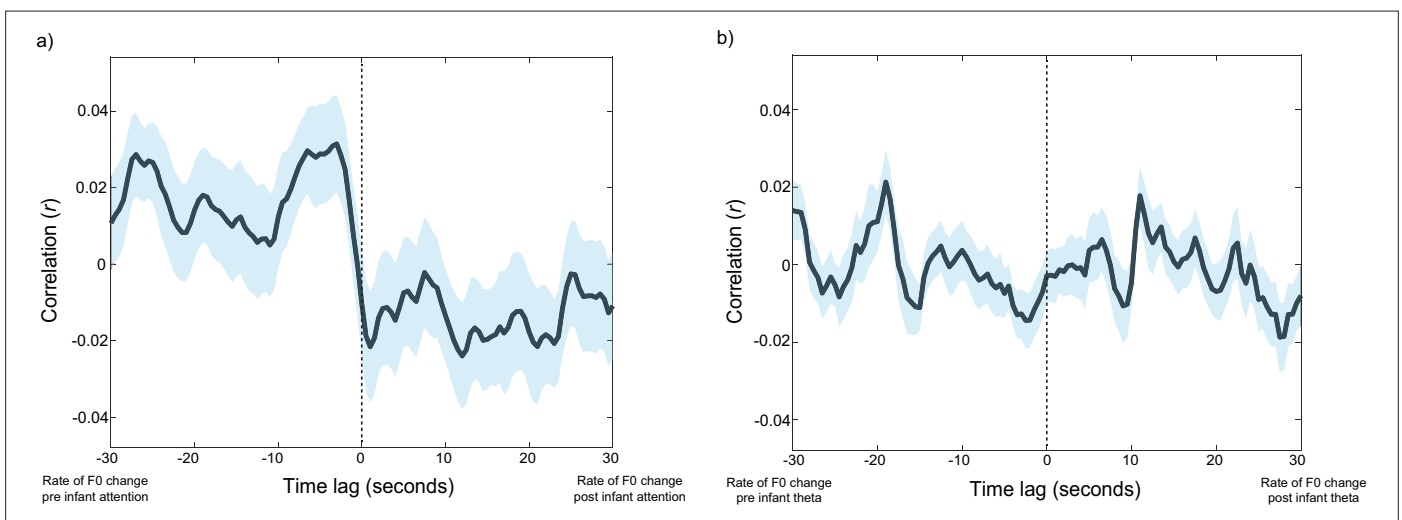

**Figure 6.** Assessing forwards-predictive associations between caregiver vocal behaviour, and infant attention, and infant theta activity. Black lines show the Pearson cross-correlation between two variables; coloured shaded areas indicate the SEM. Black horizontal lines show significant clusters of time lags (*p<0.05). (**a**) Rate of change in caregiver F0 and continuous infant attention durations (N=51). (**b**) Rate of change in caregiver F0 and infant theta activity (N=46).

the caregiver towards objects (*Figure 5e*). These analyses demonstrate immediate, reactive, change in caregiver behaviour at the onset of infant attention towards an object, as well as slower-changing modulations in their behaviour over the length of an attention episode.

### Caregiver vocal behaviour

Next, we used an identical analysis approach to examine forwards-predictive and reactive associations between infant attention and caregiver vocal behaviours. Here, we concentrate on the rate of change in F0 as a marker of auditory saliency in the caregiver's voice. In additional analyses presented in the Appendix, we also examine caregiver vocal durations, and caregiver amplitude modulations (*Appendix 1—figure 2* and *Appendix 1—figure 3*).

### Forwards-predictive associations between infant look durations and caregiver vocal behaviour

First, we computed the cross-correlations between rate of change of caregiver F0, infant attention durations, and infant endogenous neural activity. Results are shown in *Figure 6*. Cluster-based permutation analysis revealed that the time-lagged associations between infant attention and rate of change in caregiver F0 did not exceed chance (*Figure 6a*). To test whether there was any direct influence of

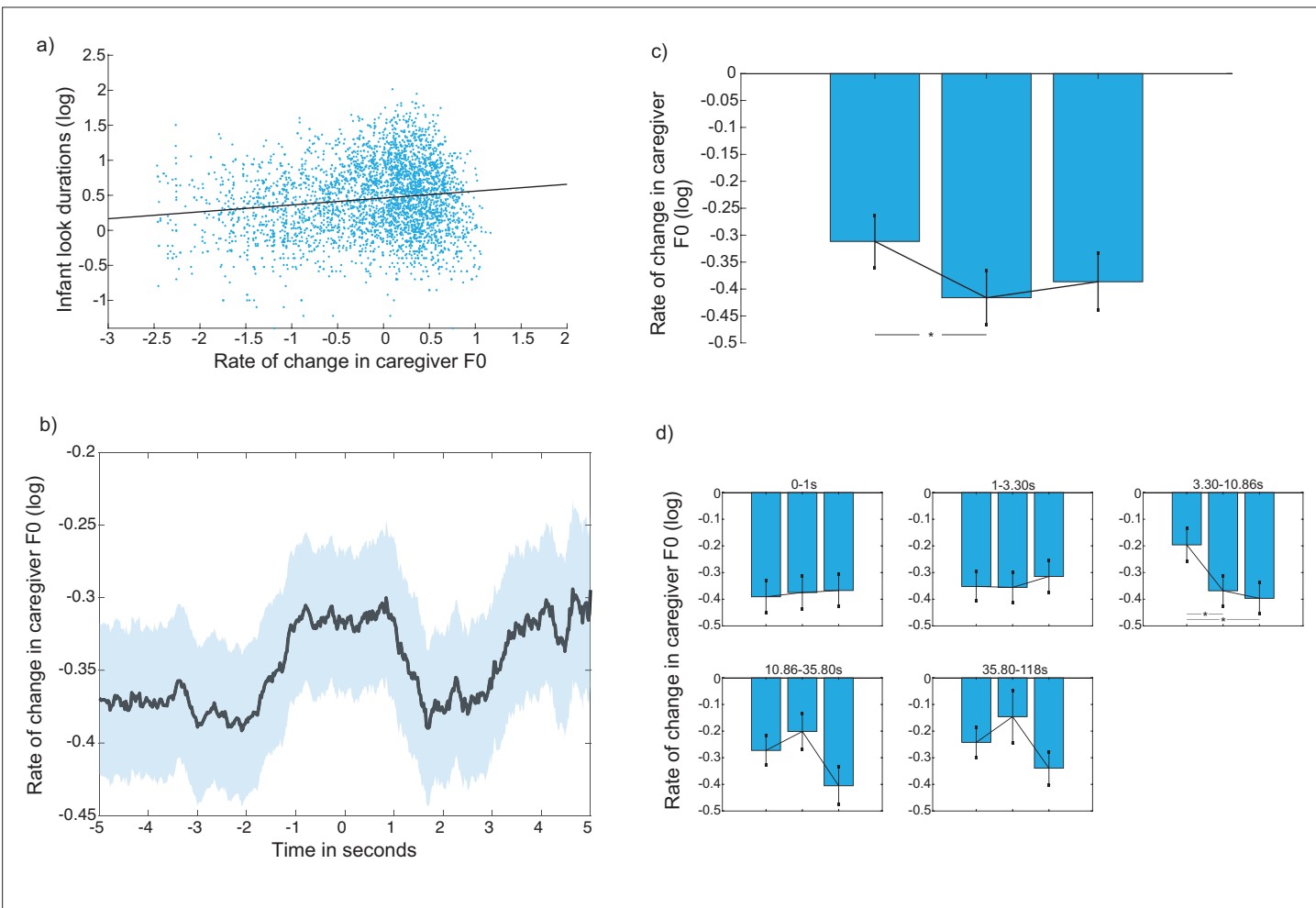

**Figure 7.** Reactive change in caregiver vocal behaviour relative to infant attention onsets. (**a**) Scatter plot of the association between infant attention durations and rate of change in caregiver F0. A linear mixed effects model revealed a significant positive association between the two variables (β=0.13; p<0.001 [Spearman's *r*=0.09, p<0.001; Pearson's *r*=0.14, p<0.001]), (**b**) Event related analysis examining reactive change in caregiver F0 in the time after the onset of an infant object look. Black line indicates the average across participants; coloured shaded area indicates SEM. (**c**) Modulation analysis: each bar shows the median for each chunk across participants; errors bars show the SEM. Wilcoxon signed ranks tests explored significant differences between attention chunks (*p<0.05), (**d**) Same as (**c**), binned by infant attention durations. N=51 for all analyses.

caregiver behaviour on modulations in infant endogenous neural activity, the same analyses were subsequently repeated relative to infant theta activity (*Figure 6b*): cluster-based permutation analysis again suggested no significant association between rate of change in caregiver F0 and infant theta activity. The same analyses are presented relative to caregiver vocal durations and amplitude modulations in *Appendix 1—figure 2*, which showed a similar pattern of results.

### Reactive change in caregiver vocal behaviour following infant look onset

To examine whether caregivers reactively adapted their vocal behaviour to changes in infant attention, we repeated the same analysis presented in section *Reactive change in caregiver look durations following infant look onset*, with rate of change of caregiver F0 as the dependent variable. The event-related analysis revealed no increase in the rate of change in caregiver F0 relative to infant attention onsets: cluster-based permutation analysis revealed no change above chance levels (*Figure 7b*). This suggests that modulations in caregiver's speech were not immediately reactive to infant attention onsets towards objects. Over the length of individual attention episodes towards objects, however, linear mixed effects models, predicting infant attention durations from caregiver F0, with participant as a random factor, revealed that longer object looks associated with a greater rate of change in caregiver F0 ($\beta$=0.13; p<0.001; *Figure 7a*), and, for looks lasting between 3 and 10 s, caregivers decreased the rate of change in the fundamental frequency of their voice, over the course of a look (*Figure 7c and d*). The exact same analysis relative to caregiver vocal durations and amplitude modulations showed a similar pattern of findings, which is presented in *Appendix 1—figure 3*.

### Summary

In summary, longer infant object look durations associated with a greater rate of change in caregiver F0, overall. Caregiver vocal behaviour showed no event-related change relative to infant attention onsets towards objects, but longer attention durations were associated with a decrease in the rate of change in F0.

## Discussion

Recording EEG activity from infants whilst they engaged in shared, naturalistic interactions with their caregiver, we examined the endogenous mechanisms and inter-personal contingencies that drive the allocation of infant attention during social interactions. To do so, we conducted three sets of analyses. First, we examined whether caregiver and infant attention patterns act as coupled oscillators (part 1). Second, we examined how infants' endogenous neural activity forwards-predicted attention durations, and how it changed reactively relative to the onsets of infant attention episodes towards objects (part 2). Third, we examined how caregiver gaze and vocal behaviour forwards-predicted infant attention durations, and how it changed reactively to the onsets of infant object looks (part 3).

When we examined whether and how endogenous cognitive processes predict infant attention, we found evidence for two distinct mechanisms. First, oscillatory mechanisms predict infant attention durations (part 1, *Figure 2*), with a period centring around 1–2 s in length. Second, independently, fluctuations in neural markers of infants' engagement or interest associated with their attention durations over the course of the interaction (part 2). Cross-correlation analyses revealed associations between infant theta activity and infant attention durations, such that increases in infant attention durations forwards-predicted increases in infant theta activity more than vice versa (*Figure 3a*). Overall, average theta power during an object-directed attention episode correlated with the duration of that episode (*Figure 3b*). Infant theta activity did not, however, show any immediate change at the beginning of an object-directed attention episode, or modulate over the length of longer attention episodes (*Figure 3c*; *Appendix 1—figure 1*). This last result may appear inconsistent with other previous findings that theta activity increases over the course of a sustained attention episode (*Xie et al., 2018*). The reason for this is likely to be methodological, as Xie and colleagues measured infants' attention while they were alone watching unfolding events on TV, while infants in our task were playing with the same toy over the course of an attention episode, and were engaged in a social interaction with their caregiver.

Overall these findings suggest, partially consistent with the predictions of active learning models (*Begus and Southgate, 2018*; *Kidd and Hayden, 2015*), that infants' own endogenous cognitive

processing is one mechanism that drives and maintains infant attention during online interactions. Strikingly, however, we found that attention durations forward-predicted theta power more than *vice versa*. Importantly, this finding replicates previous naturalistic work, where caregivers and infants engaged in joint table-top play with one toy across a table (in the current study caregivers and infants switched attention between three objects). Consistent with the findings of the current study, Wass et al. also showed that infant attention towards the object forwards-predicted increases in infant theta activity (*Wass et al., 2018a*). One possible interpretation of this finding is that longer attention durations by the infant drive incremental increases in infants' endogenous control over the allocation of their attention, through self-sustaining, bidirectional interactions between their own exploratory behaviours and information gain from the environment (*Kidd et al., 2012*; *Kidd et al., 2014*; *Oudeyer and Smith, 2016*; *Saez de Urabain et al., 2017*). Whilst infants' attention shifts may often be initiated as random behaviours (*Robertson, 2014*; *Stephens and Charnov, 1982*), at times, these self-sustaining interactions drive increases in infants' endogenous attention control, over the course of consecutive attention onsets.

Next, in order to evaluate the hypothesis that caregivers actively organise their infants' attention, we examined the association between caregiver behaviours and infant attention. Consistent with previous research (*Yu and Smith, 2016*) we found that oscillatory mechanisms govern both caregiver and infant attention durations, but that the oscillatory period of infant attention durations is longer (centring around 1–2 s in length) compared with caregivers (centring around 200–500ms in length; part 1, *Figure 2a and b*). However, when we examined whether infant and caregiver attention patterns act as coupled oscillators, which is one mechanism through which caregiver gaze behaviour might support infant gaze behaviour (*Nuthmann and Henderson, 2010a*), we found no evidence to support this (*Figure 2c*). This suggests that mechanisms of influence between infant and caregiver attention are more likely to operate as lagged, forwards- or backwards- predictive associations, as we investigated in part 2.

On the one hand, we found little to no evidence in support of the hypothesis that adult gaze and vocal behaviours forwards-predict infant attention (part 2). Against adult-led attentional structuring perspectives of early interaction, the cross-correlation analyses showed that, overall, fluctuations in infant look durations were not forwards-predicted by changes in caregiver look durations (*Figure 4a*); rather, changes in infant neural engagement largely forward-predicted changes in caregiver attention durations (*Figure 4b*). This association was likely partially mediated by the weaker and non-significant associations observed between infant attention and caregiver attention (*Figure 4a*). We also found no evidence for co-fluctuations between the rate of change of caregiver F0 (a marker of auditory salience) and infant attention durations (*Figure 6*), suggesting that increases in caregiver vocal saliency did not forward-predict changes in infant attention.

On the other hand, we did find evidence that caregivers rapidly modulated their behaviours in response to shifts in infant attention. This was particularly evident in adult gaze behaviour, where in addition to the cross-correlation findings (*Figure 4*), our event-locked analyses, including object looks only, showed that caregiver attention durations significantly decreased after the onset of a new infant attention episode (*Figure 5a*). Over the duration of longer attention episodes, however, caregiver attention durations significantly increased (*Figure 5c*), so that, overall, the linear mixed effects model revealed that longer infant object looks were associated with longer looks by the adult partner (*Figure 5d*). A series of linear mixed effects models also revealed that longer infant attention durations towards objects co-occurred with a greater rate of change in caregiver F0, as well as longer caregiver vocalisations, and an increase in caregiver amplitude (*Figure 7*; *Appendix 1—figure 3*). The modulation analysis further showed that longer infant looks (those lasting between 3 and 10 s) were associated with a decrease in the rate of change in caregiver F0 over the course of the look. In contrast to caregiver gaze behaviour, however, there was little dynamic change in caregiver vocal behaviour immediately after attention onset (*Figure 7*, *Appendix 1—figure 3*).

Overall, the caregiver behaviours we studied were largely reactive to changes in infant attention. The rapid change in caregiver gaze in response to the onset of infant attention towards objects, beginning just before attention onset, suggests that it is unlikely that caregivers are responding to active attention sharing cues produced by the infant (*Begus et al., 2014*; *Kovács et al., 2014*; *Tomasello et al., 2007*). Indeed, similar to previous micro-behavioural studies of 12-month-old infants in shared interactions (*Yu and Smith, 2013*), infants rarely looked towards their caregiver's face (*Figure 2a*),

and, in a previous analysis of this data, infants did not increase looks to their partner's face in the time before leading an episode of joint attention (*Phillips et al., 2023*). It seems therefore unlikely that the association between infant attention and fluctuations in their own endogenous cognitive processing is related to intentionally mediated forms of communication by the infant, with the goal of directly eliciting information from their caregiver (*Phillips et al., 2023*; *Yu and Smith, 2013*).

Instead, caregivers are anticipating shifts in infant attention, and, in line with an allostatic model of inter-personal interaction, 'catching' infants' attention, and monitoring their behaviour (*Yu and Smith, 2016*). This increase in the rate of caregiver behaviour after look onsets to objects could reflect dynamic up-regulatory processes that serve to maintain infant attention: though not reflected in their vocalisations; other fast-changing salient cues such as hand movements and facial affect could also increase in variability (*Meyer et al., 2023*). The down-regulation of caregiver attention over the course of longer attention episodes to objects by the infant might subsequently index decoupling of caregivers' regulatory processes from infant attention; this is also reflected in the decreased rate of change in caregiver F0 (*Figure 7c*). Combined, therefore, our findings suggest that, during interactions at the end of the first year, infant attention is structured through joint but independent influences of caregiver responsivity and regulation, and their own intrinsically motivated engagement.

In this perspective, our results can be interpreted relative to neurocomputational, associative accounts of active learning in early infancy (*Kidd and Hayden, 2015*; *Oudeyer and Smith, 2016*). These accounts postulate that contingent changes in the environment in response to actions produced by the infant improves infants' prediction and control over their own behaviour (*Friston et al., 2012*; *Friston, 2019*). In the context of shared interaction, consistent and contingent responsiveness by the caregiver to infant attention gives meaning to infants' behaviour, increasing infant engagement and further exploratory behaviours (*Oudeyer and Smith, 2016*; *Smith and Breazeal, 2007*). Over time, therefore, infants' experience of repeated interactive contingencies could influence how controlled processes begin to guide their attention, as well as their sensitivity to and engagement in intentionally mediated forms of shared communication (*Smith and Breazeal, 2007*).

This has implications for how we view and understand the interactive processes that support how infants begin to use and engage with a language system. Previous accounts have emphasised the role of the caregiver in structuring infant learning in joint attentional frames, where they use clear ostensive signals to guide infant attention, and support word-object representations (*Lieven, 2016*; *Tomasello et al., 2007*). The present study, however, found no evidence that increases in salient cues by the caregiver forward-predicted increases in infant attention durations. Increases in infant attentiveness are instead related to inter-dyadic, sensorimotor processes that are independent of the influence of infants' own endogenous cognitive process. How these fast-acting intra- and inter-individual influences on infant attention support early language acquisition should be a key focus for future research (*Yu et al., 2021*; *Yu and Smith, 2012*).

The naturalistic design of our study is a strength as well as a limitation. Of note, we were unable to control how much infants moved during the interaction, which may have contributed to eye movement artifacts time-locked to shifts in infant attention. However, eye-movement artifacts were removed using ICA decomposition, and, although this does not remove all artifact introduced to the EEG signal (*Marriott Haresign et al., 2021*), the associations observed between infant attention and theta activity suggest that this did not affect our main findings. If eye-movement artifact influenced the association between infant attention durations and theta activity, shorter attention episodes ought to associate with more theta activity, which was not the case.

In future work it will be important to take a more holistic, computational and multi-modal approach to studying how factors intrinsic to the infant, and the inter-personal behavioural contingencies of the dyad, structure infant attention and behaviour (*Xu et al., 2020*). For example, studying how inter-related multi-modal patterns of caregiver behaviour, such as body movement (*Meyer et al., 2023*), facial affect (*Murray et al., 2016a*; *Rayson et al., 2017*) and vocalisations (*Goldstein and Schwade, 2008*) support infants' engagement in joint attention, will build on the work that we reported here. In addition to the micro-dynamic analyses that we present here, it will also be important for future work to employ modelling approaches to further investigate infants' neural entrainment to the unidirectional and inter-dyadic action-generated contingencies of shared interaction (*Jessen et al., 2019*; *Jessen et al., 2021*). A particular focus of this work should be on studying the temporal latencies at

which entrainment and/or behavioural responsivity occur; utilising eye tracking methods will help with this.

In the current study, we also examined endogenous and interactive influences on infant attention by pooling across all infant attention episodes towards objects and their partner, and analysing overall trends. Whilst this approach is informative to understanding the mechanisms that drive infant attention, a key area for future work will be in examining the variability of the influences that drive more specific patterns of infant attention and behaviour. For example, infant attention episodes that occur with a pointing gesture or vocalisation might associate with greater increases in infants' endogenous cognitive processing, as well as specific modulations in the dynamics of the associations between the endogenous and interactive processes that influence the attention episodes occurring after that look (*Donnellan et al., 2020*). This work could utilise motion-tracking software to record multi-modal and temporally fine-grained patterns of infant behaviour, as well as data-driven clustering techniques, to derive specific attentional subtypes; and examine how these subtypes associate with the intra- and inter-individual dynamics of the interaction (*Yu and Smith, 2012*). Finally, a limitation of our study is that our findings might reflect a particular caregiving style (that of largely middle-class mothers living in East London), and it will be important in future research to study other populations, to investigate whether our results generalize to other populations and caregiving practices (*Keller, 2018*).

Overall, our findings suggest that infant attention in early interaction is asymmetric, related to their own endogenous cognitive processing and to consistent, reactive contingency to changes in their attention by the caregiver. Active learning strategies operate across the dyad; and are likely foundational to early language acquisition and socio-cognitive learning.

# Materials and methods
## Participants
Ninety-four caregiver-infant dyads took part in this study. The final overall sample with usable, coded, gaze data was 66 (17 infants were excluded due to recording error or equipment failure, 4 infants were excluded for fussiness and 6 infants were excluded due to poor quality EEG data, and limited coding resources). Of the infants with usable gaze data, 51 had additional vocal data (15 excluded due to recording error/equipment failure). Of those with gaze data, 60 infants had usable EEG data (a further 6 excluded due to noisy EEG data -see artifact rejection section below). All usable data sets available for each separate analysis were used in the results reported below (e.g. infants with gaze and EEG data but no vocal data are included in analyses exploring the association between infant EEG and gaze). The sample size was selected following power calculations presented in the original funding application (RPG-2018–281). The mean age of the final overall sample (n=66) was 11.18 months (SD = 1.27); 33 females, 30 males. All caregivers were female. See *Appendix 1—table 1* for further demographic information. Participants were recruited through baby groups and Childrens' Centers in the Boroughs of Newham and Tower Hamlets, as well as through online platforms such as Facebook, Twitter and Instagram. Written informed consent was obtained from all participants before taking part in the study, and consent to publish was obtained for all identifiable images used. All experimental procedures were reviewed and approved by the University of East London Ethics Committee (Approval number: ETH1819-0141).

## Experimental set-up
Parents and infants were seated facing each other on opposite sides of a 65 cm wide table. Infants were seated in a high-chair, within easy reach of the toys (see *Figure 1b*). The shared toy play comprised two sections, with a different set of toys in each section, each lasting ~5 min each. Two different sets of three small, age-appropriate toys were used in each section; this number was chosen to encourage caregiver and infant attention to move between the objects, whilst leaving the table uncluttered enough for caregiver and infant gaze behaviour to be accurately recorded cf. (*Yu and Smith, 2013*).

At the beginning of the play session, a researcher placed the toys on the table, in the same order for each participant, and asked the caregiver to play with their infant just as they would at home. Both researchers stayed behind a screen out of view of caregiver and infant, except for the short break between play sessions. The mean length of joint toy play recorded for play section 1 was 297.28 s (SD = 54.93) and 323.18 s (SD = 83.45) for play section 2.

## Equipment

EEG signals were recorded using a 32-chanel BioSemi gel-based ActiveTwo system with a sampling rate of 512 Hz with no online filtering using Actiview Software. The interaction was filmed using three Canon LEGRIA HF R806 camcorders recording at 50 fps. Parent and infant vocalisations were also recorded throughout the play session, using a ZOOM H4n Pro Handy Recorder and Sennheiner EW 112P G4-R receiver.

Two cameras faced the infant: one placed on the left of the caregiver, and one on the right (see *Figure 1b*). Cameras were placed so that the infant's gaze and the three objects placed on the table were clearly visible, as well as a side-view of the caregiver's torso and head. One camera faced the caregiver, positioned just behind the left or right side of the infant's high-chair (counter-balanced across participants). One microphone was attached to the caregiver's clothing and the other to the infant's high-chair.

Caregiver and infant cameras were synchronised to the EEG via radio frequency (RF) receiver LED boxes attached to each camera. The RF boxes simultaneously received trigger signals from a single source (computer running MATLAB) at the beginning of each play section, and concurrently emitted light impulses, visible in each camera. Microphone data was synchronised with the infants' video stream via a xylophone tone recorded in the infant camera and both microphones, which was hand identified in the recordings by trained coders. All systems were extensively tested and found to be free of latency and drift between EEG, camera and microphone to an accuracy of +/-20ms.

## Video coding

The visual attention of caregiver and infant was manually coded using custom-built MATLAB scripts that provided a zoomed-in image of parent and infant faces (see *Figure 1b*). Scripts used to code the gaze behaviours of caregivers and infants are available on Zenodo (*Wass, 2024*). Coders indicated the start frame (i.e. to the closest 20ms, at 50fps) that caregiver or infant looked to one of the three objects, to their partner, or looked away from the objects or their partner (i.e. became inattentive). Partner attention epsiodes included all looks to the partner's face; looks to any other parts of the body or the cap were coded as inattentive. Periods where the researcher was within camera frame were marked as uncodable, as well as instances where the caregiver or infant gaze was blocked or obscured by an object, or their eyes were outside the camera frame. Video coding was completed by two coders, who were trained by the first author. Inter-rater reliability analysis on 10% of coded inter-actions (conducted on either play section 1 or play section 2), dividing data into 20ms bins, indicated strong reliability between coders (kappa = 0.9 for caregiver coding and kappa = 0.8 for infant coding).

## Vocalisation coding

The onset and offset times of caregiver and infant vocalisations were identified using an automatic detector. The algorithm detected voiced segments and compared the volume and fundamental frequency detected in each recorded channel to infer the probable speaker (caregiver vs. infant). Iden-tification of the onset and offset times of the detector then underwent a secondary analysis by trained coders, who identified misidentification of utterances by the automatic decoder, as well as classifying the speaker for each vocalisation. As the decoder did not accurately identify onset and offset times of caregiver and infant during co-vocalisations, and, as these vocalisations could not be included in analyses of the spectral properties of caregiver vocalisations, these were excluded from all analyses. The mean percentage of caregiver vocalisations that were co-vocalisations was less than 20%: 19.43 (SD = 12.36; a box plot across all participants is presented in *Appendix 1—figure 4*). In a previous analysis conducted on a sub-sample of the data, we have shown that there is no significant change in infant vocalisations, relative to the onset of infant attention episodes, and their vocal beahviour did not distinguish between moments that they either led or followed their partners' attention during the interaction. It is therefore unlikely that inclusion of co-vocalisations in the current analyses would affect the main findings, time-locking caregiver vocalisations to infant attention.

## Infant EEG artifact rejection and pre-processing

A fully automatic artifact rejection procedure including ICA was adopted, following procedures from commonly used toolboxes for EEG pre-processing in adults (*Bigdely-Shamlo et al., 2015*; *Mullen, 2012*) and infants (*Debnath et al., 2020*; *Gabard-Durnam et al., 2018*), and optimised and tested for

use with our naturalistic infant EEG data (*Georgieva et al., 2020*; *Marriott Haresign et al., 2022*). This was composed of the following steps: first, EEG data were high-pass filtered at 1 Hz (FIR filter with a Hamming window applied: order 3381 and 0.25/25% transition slope, passband edge of 1 Hz and a cut-off frequency at –6 dB of 0.75 Hz). Although there is debate over the appropriateness of high pass filters when measuring ERPs (see *Widmann and Schröger, 2012*), previous work suggests that this approach obtains the best possible ICA decomposition with our data (*Dimigen, 2020*; *Marriott Haresign et al., 2021*). Second, line noise was eliminated using the EEGLAB (*Bigdely-Shamlo et al., 2015*) function *clean_line.m* (*Mullen, 2012*).

Third, the data were referenced to a robust average reference (*Bigdely-Shamlo et al., 2015*). The robust reference was obtained by rejecting channels using the EEGLAB *clean_channels.m* function with the default settings and averaging the remaining channels. Fourth, noisy channels were rejected, using the EEGLAB function *clean_channels.m.* The function input parameters 'correlation threshold' and 'noise threshold' (inputs one and two) were set at 0.7 and 3, respectively; all other input parameters were set at their default values. Fifth, the channels identified in the previous stage were interpolated back, using the EEGLAB function eeg_interp.m. Interpolation is commonly carried out either before or after ICA cleaning but, in general, has been shown to make little difference to the overall decomposition (*Delorme and Makeig, 2004*). Infants with over 21% (7) electrodes interpolated were excluded from analysis. After exclusion, the mean number of electrodes interpolated for infants was 3.37 (SD = 2.27) for play section 1, and 3 (SD = 2.16) for play section 2.

Sixth, the data were low-pass filtered at 20 Hz, again using an FIR filter with a Hamming window applied identically to the high-pass filter. Seventh, continuous data were automatically rejected in a sliding 1 s epoch based on the percentage of channels (set here at 70% of channels) that exceed 5 standard deviations of the mean channel EEG power. For example, if more than 70% of channels in each 1 s epoch exceed 5 times the standard deviation of the mean power for all channels then this epoch is marked for rejection. This step was applied very coarsely to remove only the very worst sections of data (where almost all channels were affected), which can arise during times when infants fuss or pull the caps. This step was applied at this point in the pipeline so that these sections of data were not inputted into the ICA. The mean percentage of data removed in play section 1 was 11.30 (SD = 14.97), and 6.57 (SD = 6.57) for play section 2.

Data collected from the entire course of the play session (including play section 1 and play section two, as well as two further 5-min interactions) were then concatenated and ICAs were computed on the continuous data using the EEGLAB function runica.m. After ICA rejection, data from each play section were re-split.

## Pre-processing of continuous variables

Prior to conducting our main analyses, all primary variables of interest were converted into continuous variables, in order to perform time-lagged and event-locked methods of analysis, relative to infant attention (see *Figure 1d*). All continuous variables were down sampled to match the sampling rate of the video cameras (50 Hz).

### Infant theta activity over fronto-central electrodes

First, missing data points were excluded from the continuous time-series. Where one or more of the fronto-central electrodes of an individual infant exceeded 100uV for more than 15% of the interaction, the infant's continuous theta time-series was excluded from analyses. Next, time-frequency decomposition was conducted via continuous morlet wavelet analysis to extract EEG activity occurring at frequencies ranging from 1 to 16 Hz. Specifically, the EEG signal at each channel was convolved with Gaussian-windowed complex sine-waves, ranging from 1 to 16 Hz, in linearly spaced intervals. The width of the guassian was set to 7 cycles. Power was subsequently extracted as the absolute value squared, resulting from the complex signal. After decomposition, to get rid of edge artifacts caused by convolution, the first and last 500ms of the time series were treated as missing data points. Missing data points were then re-inserted into the continuous variable as blank values, and the 500ms before and after these chunks of data also excluded. For each time point, for each frequency, power was expressed as relative power (i.e. the total power at that frequency, divided by the total power over all frequencies). EEG activity was then averaged over frequencies ranging from 3 to 6 Hz, and averaged over fronto-central electrodes (AF3, AF4, FC1, FC2, F3, F4, Fz; see *Figure 1*). This electrode cluster

was chosen based on previous infant literature (*Braithwaite et al., 2020*). This continuous, one-dimensional variable was then downsampled from 512 to 50 Hz by taking the median theta activity for every 10 samples of data, and, in each second, taking an extra 1 sample for 3 time points and an extra 2 samples for 1 time point. The spacing of these added samples was shuffled for each second of data.

## Continuous attention durations

An attention episode was defined as a discrete period of attention towards one of the play objects on the table, or to the partner. The end of each attention episode was defined as the moment where the participant first looked away from the target towards another object, towards the partner, or towards another location that was not either the object or the partner (coded as non-target attention). See *Figure 1d* for an example. Parts of the caregiver/infant gaze coded as uncodable were treated as missing data points, as well as the looks occurring in the time just before and after (in order to account for the fact that we do not know how long these looks last).

For the analyses in parts 2 and 3, which examine the associations between attention durations and other measures, we recoded each look based on the duration in seconds of that look. The durations of each look were then used to produce a continuous look duration variable, irrespective of whether that look was towards the object, partner, or non-target (see *Figure 1d*). These analyses examine therefore the associations between the durations of attention episodes and, respectively, endogenous infant neural activity (part 2) and caregiver behaviour (part 3).

## Binary attention durations

For the analyses in part 1, which examine the temporal oscillatory patterns of attention shifts, we recoded each look alternatively as a 0 or 1 from the first look of the interaction to the last (see *Figure 1d*). These analyses examine therefore the temporal inter-dependencies between attention durations (within an individual and across the dyad), irrespective of where the attention is directed.

## Rate of change in the fundamental frequency (F0) of the caregiver's voice

The fundamental frequency of the caregiver's voice was extracted using Praat (*Borsema and Weenik, 2019*), with floor and ceiling parameters set between 75 and 600 Hz. Caregiver fundamental frequency was placed into the continuous variable only where the coder had identified that section of speech as the caregiver speaking, so that infant vocalisations were not included in the analysis. Due to the caregiver being within variable distance of their microphones, some clipping was identified in a sample of the microphone recordings. A stringent clipping identification algorithm was used (see Appendix 1: clipping identification algorithm and *Appendix 1—figure 5*) to remove parts of the microphone data where clipping occurred (*Hansen et al., 2021*). Vocalisations where any clipping was identified were set to missing data points. Interactions with more than 30% missing vocalisations were excluded from the analyses. Statistics on the number of vocalisations excluded on this basis is presented in *Appendix 1—figure 5*. Co-vocalisations were set to missing data points.

Next, unvoiced sounds and periods between vocalisations were interpolated, using MATLAB's interp1 function. To reduce the likelihood of background noise (e.g. toy clacks) affecting the fundamental frequency, the interpolated F0 variable was low-pass filtered at 20 Hz using a 9th order butterworth filter. The rate of change in the caregivers' fundamental frequency was computed by taking the sum of the derivative in 1000ms intervals. The start and end points of each interval were then converted to time in camera frames, and the rate of change values inserted for the 50 corresponding frames.

See Appendix 1 for a description of the computation of caregiver vocal durations and amplitude modulations.

## Analysis procedures

### Procedures for part 1

#### Partial autocorrelation function

The partial auto-correlation function (PACF) of the caregiver and infant gaze time series was computed separately, over a range of time intervals, from 100 to 1000ms. First, the gaze time series was converted to a continuous binary variable, with either a 1 or 0 inserted into the time series for the duration of

each attention episode, alternated for each consecutive look. The PACF was then computed by fitting an ordinary least squares regression model, at time-lags ranging from 0 to 10 s, in 100ms intervals, controlling for all previous time-lags on each iteration. This analysis was repeated at intervals of 200, 500, and 1000ms.

Shuffled time series

To investigate whether the shape of the PACF reflected the temporal distribution of infant/caregiver attention episodes or more simply the frequency distribution (i.e. infant/caregiver attention episodes frequently last a similar length *Brookshire, 2022*), we conducted a permutation procedure, whereby, for each infant, their attention duration time series was shuffled randomly in time to produce a binary gaze time series of shuffled attention durations. The PACF was then computed for this time series in exactly the same way described above. This procedure was subsequently repeated 100 times for each participant, before averaging over all permutations and participants.

## Binary cross-correlation

Computation

The cross-correlation between caregiver and infant binary attention time-series was computed at lags 0 to +10 s in 500ms intervals. The zero-lagged cross-correlation was first computed between the two binary attention variables using a Spearman correlation. The infant's time series was then moved forwards in time and the Spearman correlation computed between the two time-series at each 500ms interval. The cross-correlations at each time-lag were then averaged over the two interactions for each participant, and then averaged over all participants.

Poisson baselines

Poisson baselines were created by computing time series of the Poisson point process with the length of look durations matching the average length of look durations in the actual data, for caregivers and infants separately (*Takahashi et al., 2013*). These variables were then converted to binary look duration variables, and the cross-correlation between the two binary time-series computed in exactly the same way described above. This procedure was repeated 100 times in order to create a baseline permutation distribution.

Significance testing

A cluster-based permutation approach was used to investigate whether the binary cross correlation differed significantly from the Poisson baseline distribution over any time-period. This approach controls for family-wise error rate using a non-parametric Monte Carlo method (*Maris and Oostenveld, 2007*). First, the cross-correlation at each time lag in the observed data was compared with the Poisson baseline distribution at that time lag, and values falling above the 97.5th centile and below the 2.5th centile were accepted as significant (corresponding to a significance level of 0.05). Next, to examine the likelihood of clusters of significant time points in the observed data occurring by chance, a cluster-threshold was computed using a leave-on-out procedure with the Poisson baselines. On each iteration, one baseline was compared with the 99 other baselines, and significant time-points identified using the same method described above. The largest cluster found on each iteration was identified to create a random permutation distribution of cluster sizes. The clusters identified in the observed data were then compared with this permutation distribution of maximum cluster sizes, and clusters falling above the 95th centile were considered significant (corresponding to a significance level of 0.05).

# Procedures for parts 2 and 3

## Cross-correlation analyses

Cross-correlations were computed between continuous infant attention durations, the continuous caregiver variables and infant theta activity. All analyses for the continuous caregiver variables were subsequently repeated relative to infant theta activity.

## Computation

First, the time series of each variable were log transformed, and outliers falling 2 inter-quartile ranges above the upper quartile and two inter-quartile ranges below the lower quartile removed. A detrend was then applied to each variable; linear and quadratic bivariate polynomials were fit to each transformed time-series, and the residuals of the model of best fit computed. The cross-correlation between

the two variables was then computed at lags –30 to +30 s in 500ms intervals. The zero-lagged cross-correlation was first computed between the two variables using a Pearson correlation. The caregiver's time series (or infant theta activity where this was computed relative to infant attention durations) was then moved backwards in time (to compute negative lag correlations), or forwards in time (to compute positive lag correlations), and the Pearson correlation computed between the two time-series at each 500ms interval. In this way, we estimated how the association between the two variables changed with increasing time lags. The cross-correlations at each time-lag were then averaged over the two interactions for each participant, and then averaged over all participants.

## Significance testing

A cluster-based permutation approach was used to investigate whether the time-lagged cross correlation differed significantly from chance over any time period. This approach controls for family-wise error rate using a non-parametric Monte Carlo method (*Maris and Oostenveld, 2007*). To create a random permutation distribution at each time-lag, each participant was randomly paired with another participant, through a process of derangement, and the cross-correlation between the caregiver and infant variables computed, and averaged over participants in exactly the same way described above. This procedure was then repeated 1000 times, resulting in a random permutation distribution at each time lag. Next, the cross-correlation at each time lag in the observed data was compared with the permutation distribution at that time lag, and values falling above the 97.5th centile and below the 2.5th centile were accepted as significant (corresponding to a significance level of 0.05). To examine the likelihood of clusters of significant time points in the observed data occurring by chance, a cluster-threshold was computed using a leave-on-out procedure on the permutation data. On each iteration, one permutation was compared with the 999 other permutations, and significant time-points identified using the same method described above. The largest cluster found on each iteration was identified to create a random permutation distribution of cluster sizes. The clusters identified in the observed data were then compared with this permutation distribution of maximum cluster sizes, and clusters falling above the 95th centile were considered significant (corresponding to a significance level of 0.05). See *Appendix 1—figure 6* for an example of an observed data time series plotted against its permutation distribution.

## Linear mixed effect models

Linear mixed effect models were used to investigate the association between caregiver behaviour, infant theta activity and infant attention durations. First, for each participant, for each object attention episode, the continuous caregiver behavioural variable (or infant theta activity) was averaged over the length of the infant attention episode, to obtain one value per infant attention duration. Next, each variable was log-transformed, and outliers 2 inter-quartile ranges above the upper quartile and two inter-quartile ranges below the lower quartile removed. Finally, linear mixed effects models were fitted, with caregiver behaviour, or infant theta activity as the fixed effect, and infant attention durations as the response variable, using the *fitlme* function in MATLAB. To control for between-participant variability in infants' average attention durations (*Suarez-Rivera et al., 2019*), random intercepts were specified for participants, as well as uncorrelated by-participant random slopes, to control for differences between participants in the strength of the effect of caregiver behaviours on infant look durations (*Brown, 2021*; *Gelman and Hill, 2006*).

## Attention onset event-related analysis

### Computation

Before event-locking the continuous variables to infant attention, the continuous variable (caregiver behaviour / infant theta activity) was log-transformed, and outliers removed, applying a similar procedure to that described above. First, the frame of the onset of each infant object look, as well as the duration of that look was extracted from the infant gaze time series. Next, for each continuous variable, the frames occurring five seconds before and five seconds after the onset of each infant look were extracted from the continuous time series. Given the fact that we were interested in how caregiver behaviour changed around the onset of an attention episode, where the infant shifted gaze again in the 5 s time period after attention onset, the values in the continuous caregiver variable were set to missing data points. The continuous frames occurring before and after each look were then

averaged over looks, for each interaction, resulting in an averaged continuous variable along the time dimension. These values were then averaged over interactions for each participant, before averaging over all participants.

In order to explore the possibility that the length of the infant attention episode might affect how the caregiver's behaviour changed around the onset of that episode, exactly the same analysis was repeated on attention durations of different lengths, in 5 log-spaced intervals, ranging from 0 to the longest attention episode identified across the datasets (118 s).

### Significance testing

Significance testing followed exactly the same procedures outlined in the cross-correlation analysis section.

### Modulation during attention episodes

For this analysis, all continuous data variables (caregiver behaviour / infant theta activity) were log transformed and outliers removed (see above). Then, for each infant object look, the continuous caregiver behaviour / infant theta activity was extracted over the length of that attention episode, and divided into 3 equal-spaced chunks. The continuous data variable occurring in the first half of each chunk was then averaged for each attention episode, before being averaged over all episodes for that interaction. Averaged chunks from play section 1 and play section 2 were then averaged together for each participant, and the mean over all participants, for each chunk, computed. A series of Wilcoxon Signed ranks tests assessed whether the chunks differed to each other, compared to that which would be expected by chance. The Benjamini-Hochberg false discovery rate procedure was applied to correct for multiple comparisons (p<0.05; *Benjamini and Hochberg, 1995*).

Similar to the event-related analysis, infant object look durations were divided into 5 log-spaced bins to assess whether modulations in infant endogenous cognitive processing or caregiver behaviour differed for episodes lasting different lengths: exactly the same procedure was repeated for each duration bin.

## Acknowledgements

Thank you to Dean Matthews for help with data coding. Thanks to members of the UEL BabyDev Lab for comments and discussions on earlier drafts of this manuscript, and to all participating children and caregivers. This research was funded by the Leverhulme Trust [RPG-2018–281], and the European Research Council (ERC) under the European Union's Horizon 2020 research and innovation programme (grant agreement No. [853251 - ONACSA])

## Additional information

### Funding

| Funder | Grant reference number | Author |
|---|---|---|
| Leverhulme Trust | RPG-2018-281 | Sam V Wass |
| European Research Council | 853251 - ONACSA | Sam V Wass |

The funders had no role in study design, data collection and interpretation, or the decision to submit the work for publication.

### Author contributions

Emily AM Phillips, Conceptualization, Data curation, Formal analysis, Methodology, Writing – original draft, Project administration, Writing – review and editing; Louise Goupil, Methodology, Writing – original draft, Writing – review and editing; Megan Whitehorn, Data curation, Project administration, Writing – review and editing; Emma Bruce-Gardyne, Florian A Csolsim, Navsheen Kaur, Emily Greenwood, Data curation; Ira Marriott Haresign, Data curation, Formal analysis, Project administration,

Writing – review and editing; Sam V Wass, Conceptualization, Supervision, Funding acquisition, Investigation, Methodology, Writing – review and editing

## Author ORCIDs
Emily AM Phillips ⬦ https://orcid.org/0000-0003-4244-6900
Sam V Wass ⬦ https://orcid.org/0000-0002-7421-3493

## Ethics
The project was approved by the Research Ethics Committee at the University of East London (approval number: ETH1819-0141). Informed consent, and consent to publish the photos in Figure 1 were obtained from caregivers in the usual manner.

Reviewer #2 (Public Review): https://doi.org/10.7554/eLife.88775.3.sa1
Author response https://doi.org/10.7554/eLife.88775.3.sa2

# Additional files

## Supplementary files
MDAR checklist

## Data availability
De-identified versions of the EEG and behavioural time-series data analysed for this paper are available on Dryad. Due to the personally identifiable nature of the raw data, these data files (video and microphone recordings) are not publicly accessible. Researchers who wish to access the raw data should email the lead author emily.phillips@bbk.ac.uk. Permission to access the raw data will be granted as long as the applicant can guarantee that certain privacy guidelines (e.g. storing the data only on secure, encrypted servers, and a guarantee not to share it with anyone else) can be provided. In order to allow access to the raw data the name of the applicant will also need to be added to our current ethics approval from the University of East London. This is expected to be routine, as long as the applicant is able to provide these guarantees. The code used to conduct all main analyses is available on Zenodo (*Phillips et al., 2024*).

The following dataset was generated:

| Author(s) | Year | Dataset title | Dataset URL | Database and Identifier |
|---|---|---|---|---|
| Phillips E | 2024 | Endogenous oscillatory rhythms and interactive contingencies jointly influence infant attention during early infant-caregiver interaction | https://doi.org/10.5061/dryad.2547d7x1c | Dryad Digital Repository, 10.5061/dryad.2547d7x1c |

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

# Appendix 1

## Clipping identification algorithm

The clipping algorithm was based on that outlined by *Hansen et al., 2021*. First, points in the speech signal reaching the maximum or minimum amplitude were identified. Next, to identify whether each max/min value was the beginning of a clipping event, the algorithm detected whether the value next to this point was 99.5%+/-of the max/min. A clipping event was considered to have ended where 3 consecutive values below/above the 99.5% threshold occurred. All vocalisations involving any clipping were excluded from analyses.

## Caregiver vocalisation durations

The length of each caregiver vocalisation was computed in seconds and inserted into the video-frame time series for the duration of that vocalisation. Periods where the caregiver was not vocalising (i.e. vocal pauses) were set to missing data points. Times where co-vocalisations occurred were also set to missing data points.

## Caregiver amplitude modulations

Amplitude modulations in the caregivers' speech were extracted using the NSL toolbox (*Chi et al., 2005*). First, the speech signal was downsampled to 16 kHz. The 128-channel auditory spectrogram, with centre frequencies ranging from 180 to 7246 Hz was then computed (frame length = 5ms, time constant = 8ms, no nonlinear filtering), and the band-specific envelopes summed across frequencies to obtain the broadband envelope of the speech signal. The amplitude envelope was inserted into the continuous variable only where the coder had identified the caregiver as vocalising: all vocal pauses were treated as missing data points. Clipped vocalisations were also identified using the same method described above, and set as missing values. Finally, the continuous amplitude variable was synchronised to the video frames.

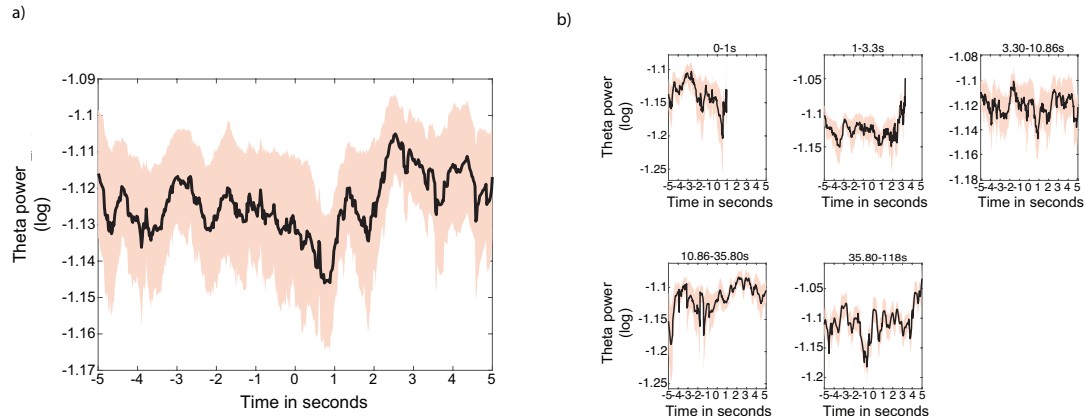

**Appendix 1—figure 1.** Event-related analysis for relative infant theta activity. (**a**) Event-related analysis showing change in infant theta activity around infant attention onsets to objects: black line shows average infant theta activity (log); coloured shaded areas indicate the SEM. Black horizontal line shows areas of significance revealed by the cluster-based permutation analysis (*p<0.05). Cluster-based permutation analysis revealed no significant clusters of time points (all p>0.05). (**b**) event-related analysis split by infant object attention-duration time bins. Black lines show average infant theta activity (log); shaded areas indicate the SEM. Black horizontal lines shows areas of significance revealed by the cluster-based permutation analysis (*p<0.05). Permutation analysis again revealed no significant clusters of time points (all p>0.05). N=60 for all analyses.

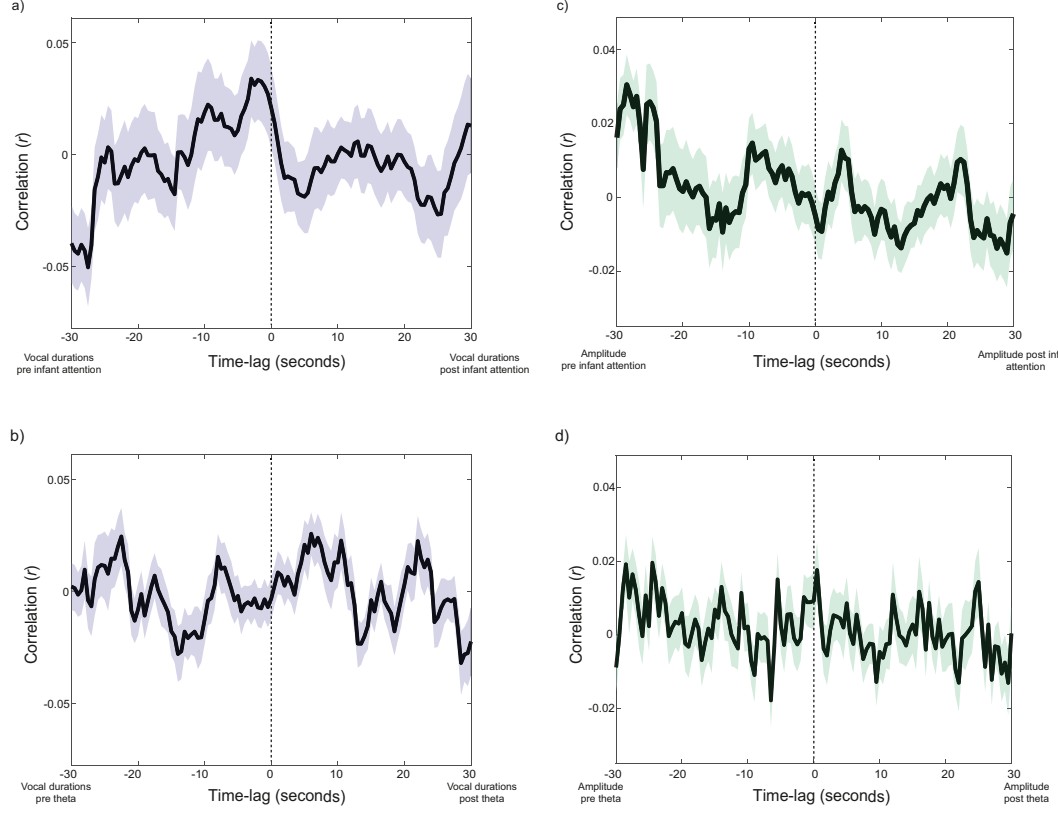

**Appendix 1—figure 2.** Assessing forwards-predictive associations between caregiver vocal behaviour, and infant attention, and infant theta activity. Black lines show the Pearson cross-correlation between two variables; shaded areas indicate the SEM. Black horizontal lines show significant clusters of time lags (*p<0.05). First column shows the association between caregiver vocal durations and (**a**) infant attention durations (N=51), and (**b**) infant theta activity (N=46). Second column shows the association between caregiver amplitude modulations (N=51) and (**c**) infant attention durations and (**d**) infant theta activity (N=46).

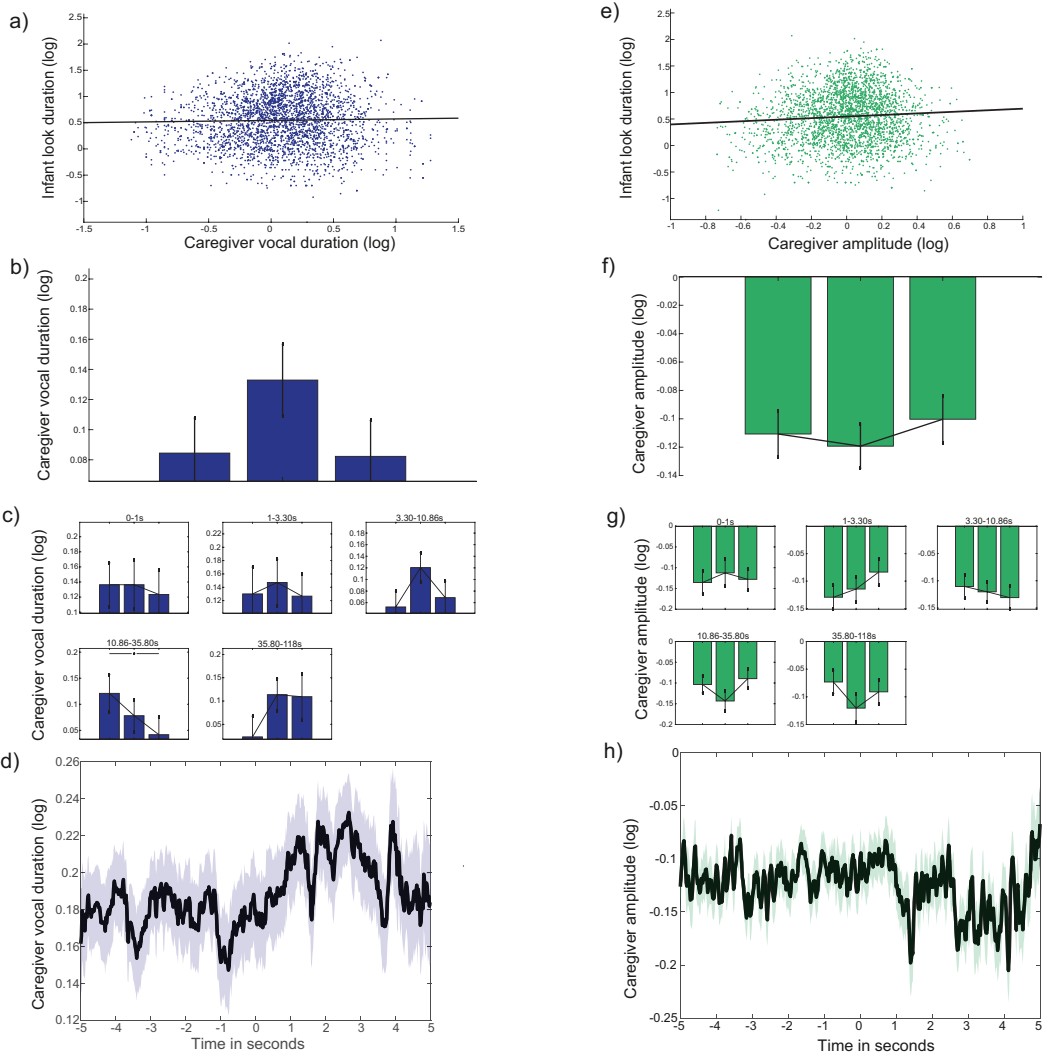

**Appendix 1—figure 3.** Reactive change in caregiver vocal behaviour relative to infant attention onsets. First row: scatter plots of the association between infant attention duration and (**a**) caregiver vocal durations, (**e**) caregiver amplitude modulations. Coloured dots show each infant object look; black line indicates linear line of best fit. Second row: modulation analysis for each infant object look for (**b**) caregiver vocal durations, (**f**) caregiver amplitude modulations. Each bar shows the median for each chunk across participants; errors bars show the SEM. Wilcoxon signed ranks tests explored significant differences between attention chunks (*p<0.05). Third row: same as second row, binned by infant attention durations, for (**c**) caregiver vocal durations, (**g**) caregiver amplitude modulations. Bottom row: event-related analysis showing change in caregiver vocal behaviour around infant attention onsets to objects: black line shows average across participants; coloured shaded areas indicate the SEM. Black horizontal lines shows areas of significance revealed by the cluster-based permutation analysis (*p<0.05). (**d**) caregiver vocal durations, (**h**) caregiver amplitude modulations. N=51 for all analyses.

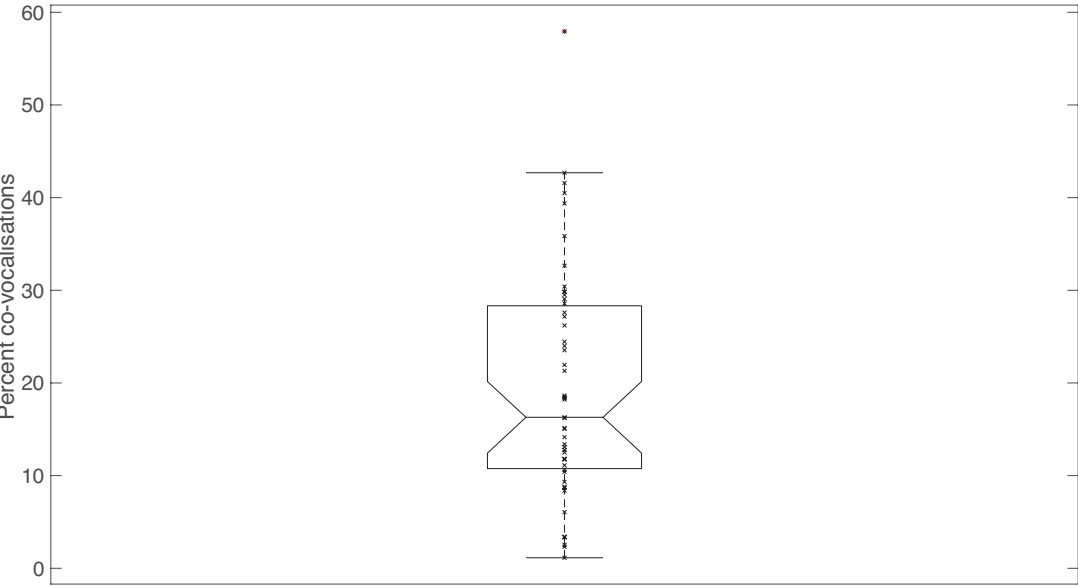

**Appendix 1—figure 4.** Box plot showing the percentage of caregiver vocalisations that were co-vocalisations across participants (N=51).

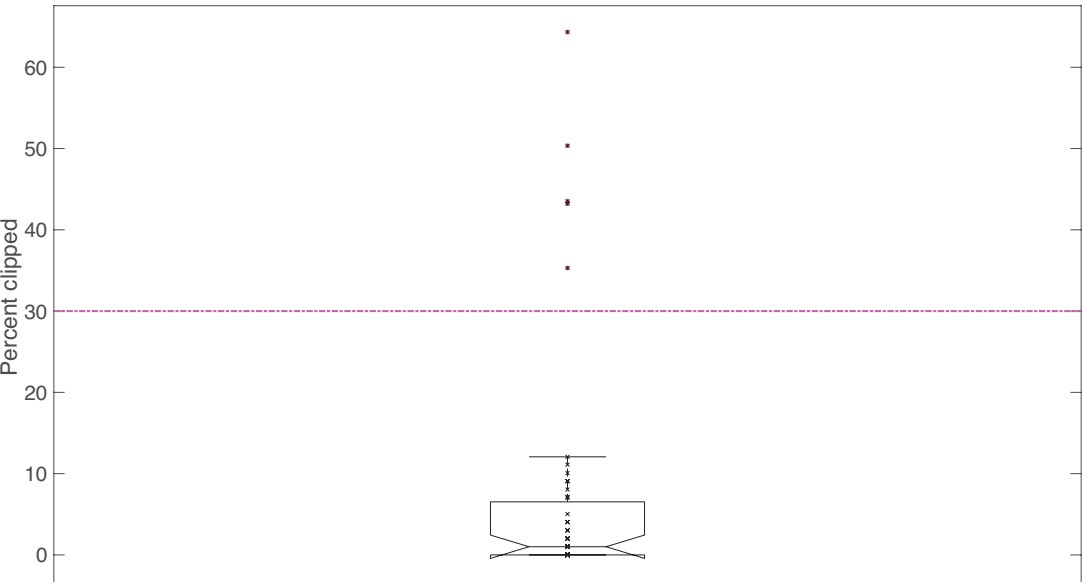

**Appendix 1—figure 5.** The box plot shows the percentage of clipped vocalisations, across participants (N=51). The pink horizontal line indicates the threshold at which participants were excluded from analyses (30%).

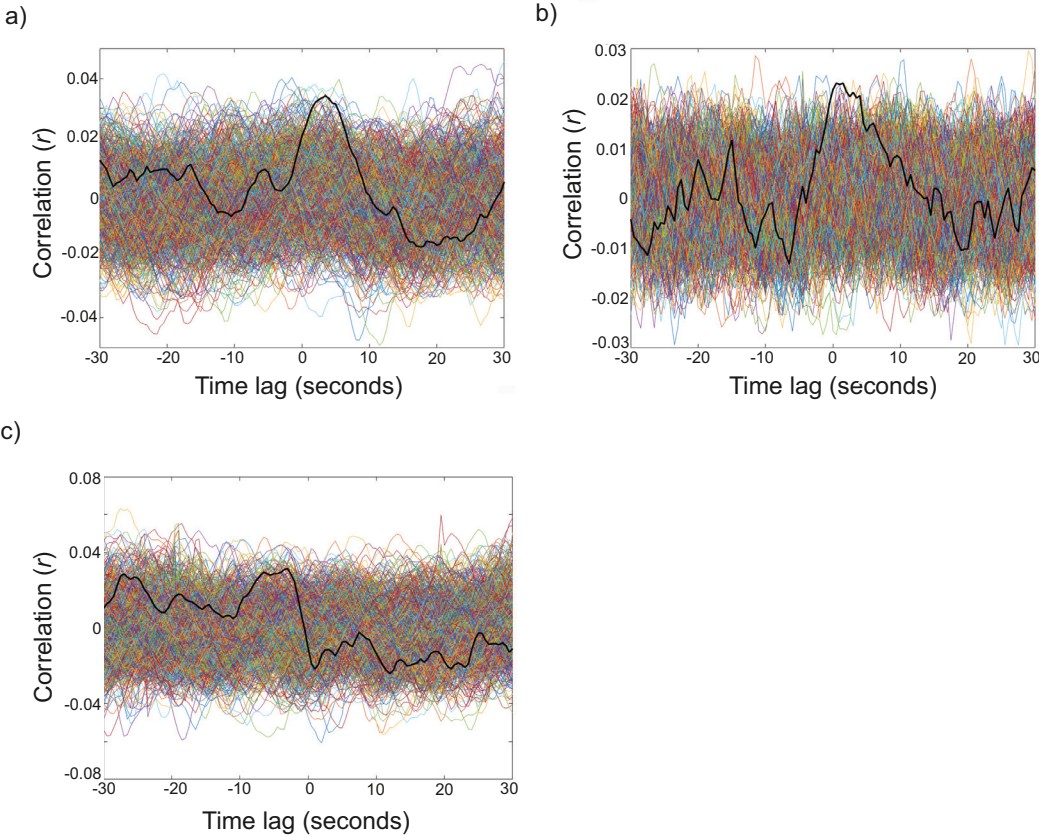

**Appendix 1—figure 6.** Observed data plotted against the permutation distribution. (**a**) infant look durations and caregiver look durations (n=66), (**b**) infant theta activity and caregiver look durations (N=60), and (**c**) infant look durations and caregiver F0 (N=51). Black lines indicate averaged, observed data values, whilst coloured lines show the average permutation values (n=1000). A significant cluster of time points is found in (**b**) only (p=0.004), whilst one cluster in (**a**) verges on significance (p=0.10).

**Appendix 1—table 1.** Demographic details for a sub-sample of the participants included in the final overall sample, for which demographic data was obtained (N=21).

| **Infant Ethnicity (%)** | **White British** | **80.95** |
|---|---|---|
| | White Other | 4.76 |
| | Mixed - White/Afro-Carib | 9.52 |
| | Other Mixed | 4.76 |
| | | |
| Household Income (%) | £26-£35 k | 4.76 |
| | £36-£50 k | 19.05 |
| | £51-£80 k | 28.57 |
| | >£80 k | 47.62 |
| | | |
| Maternal Education (%) | Postgraduate | 42.86 |
| | Undergraduate | 52.38 |
| | A-level | 4.76 |

