## [Editor Report · eLife assessment]

This study reports **important** evidence that infants' internal factors guide children's attention, and that caregivers respond to infants' attentional shifts during caregiver-infant interactions. The authors analyzed EEG data and multiple types of behaviors using **solid** methodologies that can guide future studies of neural responses during social interaction in infants.

---

## [Referee Report · Reviewer #2 (Public Review)]

Summary:

This paper acknowledges that most development occurs in social contexts, with other social partners. The authors put forth two main frameworks of how development occurs within a social interaction with a caregiver. The first, is that although social interaction with mature partners is somewhat bi-directional, mature social partners exogenously influence infant behaviors and attention through "attentional scaffolding", and that in this case infant attention is reactive to caregiver behavior. The second framework posits that caregivers support and guide infant attention by contingently responding to reorientations in infant behavior, thus caregiver behaviors are reactive to infant behavior. The aim of this paper is to use moment-to-moment analysis techniques to understand the directionality of dyadic interaction.

Strengths:

The question driving this study is interesting and a genuine gap in the literature. Almost all development occurs in the presence of a mature social partner. While it is known that these interactions are critical for development, the directionality of how these interactions unfold in real-time is less known.

The analyses are appropriate for the question at hand, capturing small moment-to-moment dynamics in both infant and child behavior, and their relationships with themselves and each other. Autocorrelations and cross-correlations are powerful tools that can uncover small but meaningful patterns in data that may not be uncovered with other more discretized analyses (i.e. regression).

Weaknesses:

While the authors improved their explanation of why they are using cross-correlations and the resting EEG patterns and what they mean, they did not address this specific piece of feedback: to explain their rationale for only focussing on fronto-temporal channels, rather than averaging channels across the whole brain.

---

## [Author Response]

The following is the authors’ response to the original reviews.

This study reports important evidence that infants' internal factors guide children's attention and that caregivers respond to infants' attentional shifts during caregiver-infant interactions. The authors analyzed EEG data and multiple types of behaviors using solid methodologies that can guide future studies of neural responses during social interaction in infants. However, the analysis is incomplete, as several methodological choices need more adequate justification.
**Reviewer #1**

**Public Review:**
The authors bring together multiple study methods (brain recordings with EEG and behavioral coding of infant and caregiver looking, and caregiver vocal changes) to understand social processes involved in infant attention. They test different hypotheses on whether caregivers scaffold attention by structuring a child's behavior, versus whether the child's attention is guided by internal factors and caregivers then respond to infants' attentional shifts. They conclude that internal processes (as measured by brain activation preceding looking) control infants' attention, and that caregivers rapidly modify their behaviors in response to changes in infant attention.The study is meticulously documented, with cutting-edge analytic approaches to testing alternative models; this type of work provides a careful and well-documented guide for how to conduct studies and process and analyze data for researchers in the relatively new area of neural response in infants in social contexts.

We are very pleased that R1 considers our work an important contribution to this developing field, and we hope that we have now addressed their concerns below.

Some concerns arise around the use of terms (for example, an infant may "look" at an object, but that does not mean the infant is actually "attending); collapsing of different types of looks (to people and objects), and the averaging of data across infants that may mask some of the individual patterns.

We thank the reviewer for this feedback and their related comments below, and we feel that our manuscript is much stronger as a result of the changes we have made. Please see blow for a detailed description of our rationale for defining and analysing the attention data, as well as the textual changes made in response to the author’s comments.

**Recommendations For The Authors**
This paper is rigorous in method, theoretically grounded, and makes an important contribution to understanding processes of infant attention, brain activity, and the reciprocal temporal features of caregiver-infant interactions. The alternative hypothesis approach sets up the questions well (although authors should temper any wording that suggests attention processes are one or the other. That is, certain bouts of infant attention can be guided by exogenous factors such as social input, and others be endogenous; so averaging across all bouts can actually mask the variation in these patterns). I appreciated the focus on multiple types of behavior (e.g., gaze, vocal fluctuations in maternal speech); the emphasis on contingent responding; and the very clear summaries of takeaways after each section. Furthermore, methods and analyses are well described, details on data processing and so on are very thorough, and visualizations aptly facilitate data interpretation. However, I am not an expert on infant neural responses in EEG and assume that a reviewer with such expertise will weigh in on the treatment and quality of the data; therefore, my comments should be interpreted in light of this lack of knowledge.

We thank R1 for these very positive and insightful comments on our analyses which are the result of a number of years of methodological and technical developmental work.

We do agree with R1 that we should more carefully word parts of our argument in the Introduction to make clear the fact that shifts in infant attention could be driven by a combination of interactive and endogenous influences. As a result of this comment, we have made direct changes to parts of the Introduction; removing any wording that suggests that these processes are ‘alternative’ or ‘separate’, and our overall aim states: ‘Here, recording EEG from infants during naturalistic interactions with their caregiver, we examined the (inter)-dependent influences of infants’ endogenous oscillatory neural activity, and inter-dyadic behavioural contingencies in organising infant attention’.

Examining variability between infant attention episodes in the factors that influence the length and timing of the attention episode is an important area for future investigation. We now include a discussion on this on page 38 of the Discussion section, with suggestions for how this could be examined. Investigating different subtypes of infant attention is methodologically challenging, given the number of infant behaviours that would need to inform such an analysis- all of which are time consuming to code. Developing automated methods for performing these kinds of analyses is an important avenue for future work.

Here, I review various issues that require revision or elaboration based on my reading of what I consider to otherwise be a solid and important research paper.Problem in the use of the term attention scaffolding. Although there may be literature precedent in the use of this term, it is problematic to narrowly define scaffolding as mother-initiated guidance of attention. A mother who responds to infant behaviors, but expands on the topic or supports continued attention, and so on, is scaffolding learning to a higher level. I would think about a different term because it currently implies a caregiver as either scaffolding OR responding contingently. It is not an either-or situation in conceptual meaning. In fact, research on social contingency (or contingent responsiveness), often views the follow-in responding as a way to scaffold learning in an infant.

Yes, we agree with R1 that the term ‘attention scaffolding’ could be confusing given the use of this term in previous work conducted with children and their caregivers in problem-solving tasks, that emphasise modulations in caregiver behaviour as a function of infant behaviour. As a result of this suggestion, we have made direct edits to the text throughout, replacing the term attentional scaffold with terms such as ‘organise’ and ‘structure’ in relation to the caregiver-leading or ‘didactic’ perspective, and terms such as ‘contingent responding’ and ‘dynamic modulation’ in relation to the caregiver-following perspective. We feel that this has much improved the clarity of the argument in the Introduction and Discussion sections.

Do individual data support the group average trends? My concern with unobservable (by definition) is that EEG data averages may mask what's going on in individual brain response. Effects appear to be small as well, which occurs in such conditions of averaging across perhaps very variable response patterns. In the interest of full transparency and open science, how many infants show the type of pattern revealed by the average graph (e.g., do neural markers of infant engagement forward predict attention for all babies? Majority?). Non-parametric tests on how many babies show a claimed pattern would offer the litmus test of significance on whether the phenomenon is robust across infants or pulled by a few infants with certain patterns of data. Ditto for all data. This would bolster my confidence in the summaries of what is going on in the infant brain. (The same applies as I suggest to attention bouts. To what extent does the forward-predict or backward-predict pattern work for all bouts, only some bouts, etc.?). I recognize that to obtain power, summaries are needed across infants and bouts, but I want to know if what's being observed is systematic.

We thank R1 for this comment and understand their concern that the overall pattern of findings reported in relation to the infants’ EEG data might obscure inter-individual variability in the associations between attention and theta power. Averaging across individual participant EEG responses is, however, the gold standard way to perform both event-locked (Jones et al., 2020) and continuous methods (Attaheri et al., 2020) of EEG analysis that are reported in the current manuscript. EEG data, and, in particular, naturalistic EEG data is inherently noisy, and averaging across participants increases the signal to noise ratio (i.e. inconsistent, and, therefore, non-task-related activity is averaged out of the response (Cohen, 2014; Noreika et al., 2020)). Examining individual EEG responses is unlikely to tell us anything meaningful, given that, if a response is not found for a particular participant, then it could be that the response is not present for that participant, or that it is present, but the EEG recording for that participant is too noisy to show the effect. Computing group-level effects, as is most common in all neuroimaging analyses, is, therefore, most optimal to examining our main research questions.

The findings reported in this analysis also replicate previous work conducted by our lab which showed that infant attention to objects significantly forward-predicted increases in infant theta activity during joint table-top play with their caregiver, involving one toy object (compared to our paradigm which involved 3;Wass et al., 2018). More recent work conducted by our lab has also shown continuous and time-locked associations between infant look durations and infant theta activity when infants play with objects on their own (Perapoch Amadó et al., 2023). To reassure readers of the replicability of the current findings, we now reference the Wass et al. (2018) study at the beginning of the Discussion section.

Could activity artifacts lead to certain reported trends? Babies typically look at an object before they touch or manipulate the object, and so longer bouts of attention likely involve a look and then a touch for lengthier time frames. If active involvement with an object (touching for example) amplifies theta activity, that may explain why attention duration forward predicts theta power. That is, baby looks, then touches, then theta activates, and coding would show visual gaze preceding the theta activation. Careful alignment of infants' touches and other such behaviors with the theta peak might help address this question, again to lend confidence to the robustness of the interpretation.

Yes, again this is a very important point, and the removal of movement-related artifact is something we have given careful attention to in the analysis of our naturalistic EEG data (Georgieva et al., 2020; Marriott Haresign et al., 2021). As a result of this comment we have made direct changes to the Results section on page 18 to more clearly signal the reader to our EEG pre-processing section before presenting the results of the cross-correlation analyses.

As we describe in the Methods section of the main text, movement-related artifacts are removed from the data with ICA decomposition, utilising an automatic-rejection algorithm, specially designed for work with our naturalistic EEG data (Marriott Haresign et al., 2021). Given that ICA rejection does not remove all artifact introduced to the EEG signal, additional analysis steps were taken to reduce the possibility that movement artifacts influenced the results of the reported analyses. As explained in the Methods section, rather than absolute theta power, relative theta was used in all EEG analyses, computed by dividing the power at each theta frequency by the summed power across all frequencies. Eye and head movement-related artifacts most often associate with broadband increases in power in the EEG signal (Cohen, 2014): computing relative theta activity therefore further reduces the potential influence of artifact on the EEG signal.

It is also important to highlight that previous work examining movement artifacts in controlled paradigms with infants has shown that limb movements actually associate with a decrease in power at theta frequencies, compared to rest (Georgieva et al., 2020). It is therefore unlikely that limb movement artifacts explain the pattern of association observed between theta power and infant attention in the current study.

That said, examining the association between body movements and fluctuations in EEG activity during naturalistic interactions is an important next step, and something our lab is currently working on. Given that touching an object is most often the end-state of a larger body movement, aligning the EEG signal to the onset of infant touch is not all that informative to understanding how body movements associate with increases and decreases in power in the EEG signal. Our lab is currently working on developing new methods using motion tracking software and arousal composites to understand how data-derived behavioural sub-types associate with differential patterns of EEG activity.

The term attention may be misleading. The behavior being examined is infant gaze or looks, with the assumption that gaze is a marker of "attention". The authors are aware that gaze can be a blank stare that doesn't reflect underlying true "attention". I recommend substitution of a conservative, more precise term that captures the variable being measured (gaze); it would then be fine to state that in their interpretation, gaze taken as a marker for attention or something like that. At minimum, using term "visual attention" can be a solution if authors do not want to use the precise term gaze. As an example, the sentence "An attention episode was defined as a discrete period of attention towards one of the play objects on the table, or to the partner" should be modified to defined as looking at a play object or partner.

We thank the reviewer for this comment, and we understand their concern with the use of the term ‘attention’ where we are referring to shifts in infant eye gaze. However, the use of this term to describe patterns of infant gaze, irrespective of whether they are ‘actually attending’ or not is used widely in the literature, in both interactive (e.g. Yu et al., 2021) and screen-based experiments examining infant attention (Richards, 2010). We therefore feel that its use in our current manuscript is acceptable and consistent with the reporting of similar interaction findings. On page 39 of the Discussion we now also include a discussion on how future research might further investigate differential subtypes of infant looks to distinguish between moments where infants are attending vs. just looking.

Why collapse across gaze to object vs. other? Conceptually, it's unclear why the same hypotheses and research questions on neural-attention (i.e., gaze in actuality) links would apply to looks to a mom's face or to an object. Some rationale would be useful to the reader as to why these two distinct behaviors are taken as following the same principles in ordering of brain and behavior. Perhaps I missed something, however, because later in the Discussion the authors state that "fluctuations in neural markers of infants' engagement or interest forward-predict their attentiveness towards objects", which suggests there was an object-focused variable only? Please clarify. (Again, sorry if I missed something).

This is a really important point, and we agree with R1 that it could have been more clearly expressed in our original submission – for which, we apologise. In the cross-correlation analyses conducted in parts 2 and 3 which examines forwards-predictive associations between infant attention durations and infant endogenous oscillatory activity (part 2), and caregiver behaviour (part 3), as R1 describes, we include all infant looks towards objects and their partner. Including all infant look types is necessary to produce a continuous variable to cross-correlate with the other continuous variables (e.g. theta activity, caregiver vocal behaviours), and, therefore, does not concentrate only on infant attention episodes towards objects.

We take the reviewers’ point that different attention and neural mechanisms may be associated with looks towards objects vs. the partner, which we now acknowledge directly on page 10 of the Introduction. However, our focus here is on the endogenous and interactive mechanisms that drive fluctuations in infant engagement with the ongoing, free-flowing interaction. Indeed, previous work has shown increases in theta activity during sustained episodes of infant attention to a range of different stimuli, including cartoon videos (Xie et al., 2018), real-life screen-based interactions (Jones et al., 2020), as well as objects (Begus et al., 2016). In the second half of part 2, we go on to address the endogenous processes that support infant attention episodes specifically towards objects.

As a result of this comment, we have made direct changes to the Introduction on page 10 to more clearly explain the looking behaviours included in the cross-correlation analysis, and the rationale behind the analysis being conducted in this way – which is different to the reactive analyses conducted in the second half of parts one and three, which examines infant object looks only. Direct edits to the text have also been made throughout the Results and Methods sections as a result of this comment, to more clearly specify the types of looks included in each analysis. Now, where we discuss the cross-correlation analyses we refer only to infant ‘attention durations’ or infant ‘attention’, whilst ‘object-directed attention’ and ‘looks towards objects’ is clearly specified in sections discussing the reactive analyses conducted in parts 2 and 3. We have also amended the Discussion on page 31so that the cross-correlation analyses is interpreted relative to infant overall attention, rather than their attention towards objects only.

Why are mothers' gazes shorter than infants' gazes? This was the flip of what I'd expect, so some interpretation would be useful to understanding the data.

This is a really interesting observation. Our findings of the looking behaviour of caregivers and infants in our joint play interactions actually correspond to much previous micro-dynamic analysis of caregiver and infant looking behaviour during early table-top interactions (Abney et al., 2017; Perapoch Amadó et al., 2023; Yu & Smith, 2013, 2016). The reason for the shorter look durations in the adult is due to the fact that the caregivers alternate their gaze between their infant and the objects (i.e. they spend a lot of the interaction time monitoring their infants’ behaviours). This can be seen in Figure 2 (see main text) which shows that caregiver looks are divided between looks to their infants and looks towards objects. In comparison, infants spend most of their time focussing on objects (see Figure 2, main text), with relatively infrequent looks to their caregiver. As a result, infant looks are, overall, longer in comparison to their caregivers’.

Minor pointsUse the term association or relation (relationships is for interpersonal relationships, not in statistics).

This has now been amended throughout.

I'm unsure I'd call the interactions "naturalistic" when they occur at a table, with select toys, EEG caps on partners, and so on. The term seems more appropriate for studies with fewer constraints that occur (for example) in a home environment, etc.

We understand R1s concern with our use of the term ‘naturalistic’ to refer to the joint play interactions that we analyse in the current study. However, we feel the term is appropriate, given that the interactions are unstructured: the only instruction given to caregivers at the beginning of the interaction is to play with their infants in the way that they might do at home. The interactions, therefore, measure free-flowing caregiver and infant behaviours, where modulations in each individual’s behaviour are the result of the intra- and inter-individual dynamics of the social exchange. This is in comparison to previous work on early infant attention development which has used more structured designs, and modulations in infant behaviour occur as a result of the parameters of the experimental task.

**Reviewer #2**

**Public Review**
Summary:This paper acknowledges that most development occurs in social contexts, with other social partners. The authors put forth two main frameworks of how development occurs within a social interaction with a caregiver. The first is that although social interaction with mature partners is somewhat bi-directional, mature social partners exogenously influence infant behaviors and attention through "attentional scaffolding", and that in this case infant attention is reactive to caregiver behavior. The second framework posits that caregivers support and guide infant attention by contingently responding to reorientations in infant behavior, thus caregiver behaviors are reactive to infant behavior. The aim of this paper is to use moment-to-moment analysis techniques to understand the directionality of dyadic interaction. It is difficult to determine whether the authors prove their point as the results are not clearly explained as is the motivation for the chosen methods.StrengthsThe question driving this study is interesting and a genuine gap in the literature. Almost all development occurs in the presence of a mature social partner. While it is known that these interactions are critical for development, the directionality of how these interactions unfold in real-time is less known.The analyses largely seem to be appropriate for the question at hand, capturing small moment-to-moment dynamics in both infant and child behavior, and their relationships with themselves and each other. Autocorrelations and cross-correlations are powerful tools that can uncover small but meaningful patterns in data that may not be uncovered with other more discretized analyses (i.e. regression).

We are pleased that R2 finds our work to be an interesting contribution to the field, which utilises appropriate analysis techniques.

WeaknessesThe major weakness of this paper is that the reader is assumed to understand why these results lead to their claimed findings. The authors need to describe more carefully their reasoning and justification for their analyses and what they hope to show. While a handful of experts would understand why autocorrelations and cross-correlations should be used, they are by no means basic analyses. It would also be helpful to use simulated data or even a simple figure to help the reader more easily understand what a significant result looks like versus an insignificant result.

We thank the reviewer for this comment, and we agree that much more detail should be added to the Introduction section. As a result of this comment, we have made direct changes to the Introduction on pages 9-11 to more clearly detail these analysis methods, our rationale for using these methods; and how we expect the results to further our understanding of the drivers of infant attention in naturalistic social interactions.

We also provide a figure in the SM (Fig. S6) to help the reader more clearly understand the permutation method used in our statistical analyses described in the Methods, on page 51, which depicts significant vs. insignificant patterns of results against their permutation distribution.

While the overall question is interesting the introduction does not properly set up the rest of the paper. The authors spend a lot of time talking about oscillatory patterns in general but leave very little discussion to the fact they are using EEG to measure these patterns. The justification for using EEG is also not very well developed. Why did the authors single out fronto-temporal channels instead of using whole brain techniques, which are more standard in the field? This is idiosyncratic and not common.

We very much agree with R2 that the rationale and justification for using EEG to understand the processes that influence infants’ attention patterns is under-developed in the current manuscript. As a result of this comment we have made direct edits to the Introduction section of the main text on pages 7-8 to more clearly describe the rationale for examining the relationship between infant EEG activity and their attention during the play interactions with their caregivers.

As we describe in the Introduction section, previous behavioural work conducted with infants has suggested that endogenous cognitive processes (i.e. fluctuations in top-down cognitive control) might be important in explaining how infants allocate their attention during free-flowing, naturalistic interactions towards the end of the first year. Oscillatory neural activity occurring at theta frequencies (3-6Hz), which can be measured with EEG, has previously been associated with top-down intrinsically guided attentional processes in both adulthood and infancy (Jones et al., 2020; Orekhova, 1999; Xie et al., 2018). Measuring fluctuations in infant theta activity therefore provides a method to examine how endogenous cognitive processes structure infant attention in naturalistic social interactions which might be otherwise unobservable behaviourally.

It is important to note that the Introduction distinguishes between two different oscillatory mechanisms that could possibly explain the organisation of infant attention over the course of the interaction. The first refers to oscillatory patterns of attention, that is, consistent attention durations produced by infants that likely reflect automatic, regulatory functions, related to fluctuations in infant arousal. The second mechanism is oscillatory neural activity occurring at theta frequencies, recorded with EEG, which, as mentioned above, is thought to reflect fluctuations in intrinsically guided attention in early infancy. We have amended the Introduction to make the distinction between the two more clear.

A worrisome weakness is that the figures are not consistently formatted. The y-axes are not consistent within figures making the data difficult to compare and interpret. Labels are also not consistent and very often the text size is way too small making reading the axes difficult. This is a noticeable lack of attention to detail.

This has now been adjusted throughout, where appropriate.

No data is provided to reproduce the figures. This does not need to include the original videos but rather the processed and de-identified data used to generate the figures. Providing the data to support reproducibility is increasingly common in the field of developmental science and the authors are greatly encouraged to do so.

This will be provided with the final manuscript.

Minor WeaknessesFigure 4, how is the pattern in a not significant while in b a very similar pattern with the same magnitude of change is? This seems like a spurious result.

The statistical analysis conducted for all cross-correlation analyses reported follows a rigorous and stringent permutation-based temporal clustering method which controls for family-wise error rate using a non-parametric Monte Carlo method (see Methods in the main text for more detail). Permutations are created by shuffling data sets between participants and, therefore, patterns of significance identified by the cluster-based permutation analysis will depend on the mean and standard deviation of the cross-correlations in the permutation distribution. Fig. S6 now depicts the cross-correlations against their permutation distributions which should help readers to understand the patterns of significance reported in the main text.

The correlations appear very weak in Figures 3b, 5a, 7e. Despite a linear mixed effects model showing a relationship, it is difficult to believe looking at the data. Both the Spearman and Pearson correlations for these plots should be clearly included in the text, figure, or figure legend.

We thank the reviewer for this comment, and agree that reporting the correlations for these plots would strengthen the findings of the linear mixed effects models reported in text. As a result, we have added both Spearman and Pearson correlations to the legends of Figures 3b, 5a and 7e, corresponding to the statistically significant relationships examined in the linear mixed effects models. The strength of the relationships are entirely consistent with those documented in other previous research that used similar methods (e.g. Piazza et al., 2018). How strong the relationship looks to the observer is entirely dependent on the graphical representation chosen to represent it. We have chosen to present the data in this way because we feel that it is the most honest way to represent the statistically significant, and very carefully analysed, effects that we have observed in our data.

Linear mixed effects models need more detail. Why were they built the way they were built? I would have appreciated seeing multiple models in the supplementary methods and a reasoning to have landed on one. There are multiple ways I can see this model being built (especially with the addition of a random intercept). Also, there are methods to test significance between models and aid in selection. That being said, although participant identity is a very common random effect, its use should be clearly stated in the main text.

We very much agree with R2 that the reporting of the linear mixed effects models needs more detail and this has now been added to the Method section (page 54). Whilst it is true that there are multiple ways in which this model could be built, given the specificity of our research questions, regarding the reactive changes in infant theta activity and caregiver behaviours that occur after infant look onsets towards objects (see pages 9-11 of the Introduction), we take a hypothesis driven approach to building the linear mixed effects models. As a result, random intercepts are specified for participants, as well as uncorrelated by-participant random slopes (Brown, 2021; Gelman & Hill, 2006; Suarez-Rivera et al., 2019). In this way, infant look durations are predicted from caregiver behaviours (or infant theta activity), controlling for between participant variability in look durations, as well as the strength of the effect of caregiver behaviours (or infant theta activity) on infant look durations.

Some parentheses aren't closed, a more careful re-reading focusing on these minor textual issues is warranted.

This has now been corrected.

Analysis of F0 seems unnecessarily complex. Is there a reason for this?

Computation of the continuous caregiver F0 variable may seem complex but we feel that all analysis steps are necessary to accurately and reliably compute this variable in our naturalistic, noisy and free-flowing interaction data. For example, we place the F0 only into segments of the interaction identified as the mum speaking so that background noises and infant vocalisations are not included in the continuous variable. We then interpolate through unvoiced segments (similar to Räsänen et al., 2018), and compute the derivative in 1000ms intervals as a measure of the rate of change. The steps taken to compute this variable have been both carefully and thoughtfully selected given the many ways in which this continuous rate of change variable could be computed (cf. Piazza et al., 2018; Räsänen et al., 2018).

The choice of a 20hz filter seems odd when an example of toy clacks is given. Toy clacks are much higher than 20hz, and a 20hz filter probably wouldn't do anything against toy clacks given that the authors already set floor and ceiling parameters of 75-600Hz in their F0 extraction.

We thank the reviewer for this comment and we can see that this part of the description of the F0 computation is confusing. A 20Hz low pass filter is applied to the data stream after extracting the F0 with floor and ceiling parameters set between 75-600Hz. The 20Hz filter therefore filters modulations in the caregivers’ F0 that occur at a modulation frequency greater than 20Hz. The 20Hz filter does not, therefore, refer to the spectral filtering of the speech signal. The description of this variable has been rephrased on page 48 of the main text.

Linear interpolation is a choice I would not have made. Where there is no data, there is no data. It feels inappropriate to assume that the data in between is simply a linear interpolation of surrounding points.

The choice to interpolate where there was no data was something we considered in a lot of detail, given the many options for dealing with missing data points in this analysis, and the difficulties involved with extracting a continuous F0 variable in our naturalistic data sets. As R2 points out, one option would be to set data points to NaN values where no F0 is detected and/ or the Mum is not vocalising. A second option, however, would be to set the continuous variable to 0s where no F0 is detected and/ or the Mum is not vocalising (where the mum is not producing sound there is no F0 so rather than setting the variable to missing data points, really it makes most objective sense to set to 0).

Either of these options (setting parts where no F0 is detected to NaN or 0) makes it difficult to then meaningfully compute the rate of change in F0: where NaN values are inserted, this reduces the number of data points in each time window; where 0s are inserted this creates large and unreal changes in F0. Inserting NaN values into the continuous variable also reduces the number of data points included in the cross-correlation and event-locked analyses. It is important to note that, in our naturalistic interactions, caregivers’ vocal patterns are characterised by lots of short vocalisations interspersed by short pauses (Phillips et al., in prep), similar to previous findings in naturalistic settings (Gratier et al., 2015). Interpolation will, therefore, have largely interpolated through the small pauses in the caregiver’s vocalisations.

The only limitation listed was related to the demographics of the sample, namely saying that middle class moms in east London. Given that the demographics of London, even east London are quite varied, it's disappointing their sample does not reflect the community they are in.

Yes we very much agree with R2 that the lack of inclusion of caregivers from wider demographic backgrounds is disappointing, and something which is often a problem in developmental research. Our lab is currently working to collect similar data from infants with a family history of ADHD, as part of a longitudinal, ongoing project, involving families from across the UK, from much more varied demographic backgrounds. We hope that the findings reported here will feed directly into the work conducted as part of this new project.

That said, demographic table of the subjects included in this study should be added.

This is now included in the SM, and referenced in the main text.

References

Abney, D. H., Warlaumont, A. S., Oller, D. K., Wallot, S., & Kello, C. T. (2017). Multiple Coordination Patterns in Infant and Adult Vocalizations. Infancy, 22(4), 514–539. https://doi.org/10.1111/infa.12165

Attaheri, A., Choisdealbha, Á. N., Di Liberto, G. M., Rocha, S., Brusini, P., Mead, N., Olawole-Scott, H., Boutris, P., Gibbon, S., Williams, I., Grey, C., Flanagan, S., & Goswami, U. (2020). Delta- and theta-band cortical tracking and phase-amplitude coupling to sung speech by infants [Preprint]. Neuroscience. https://doi.org/10.1101/2020.10.12.329326

Begus, K., Gliga, T., & Southgate, V. (2016). Infants’ preferences for native speakers are associated with an expectation of information. Proceedings of the National Academy of Sciences, 113(44), 12397–12402. https://doi.org/10.1073/pnas.1603261113

Brown, V. A. (2021). An Introduction to Linear Mixed-Effects Modeling in R.

Cohen, M. X. (2014). Analyzing neural time series data: Theory and practice. The MIT Press.

Gelman, A., & Hill, J. (2006). In Data Analysis using Regression and mulilevel/Hierachical Models. Cambridge University Press.

Georgieva, S., Lester, S., Noreika, V., Yilmaz, M. N., Wass, S., & Leong, V. (2020). Toward the Understanding of Topographical and Spectral Signatures of Infant Movement Artifacts in Naturalistic EEG. Frontiers in Neuroscience, 14, 352. https://doi.org/10.3389/fnins.2020.00352

Gratier, M., Devouche, E., Guellai, B., Infanti, R., Yilmaz, E., & Parlato-Oliveira, E. (2015). Early development of turn-taking in vocal interaction between mothers and infants. Frontiers in Psychology, 6. https://doi.org/10.3389/fpsyg.2015.01167

Jones, E. J. H., Goodwin, A., Orekhova, E., Charman, T., Dawson, G., Webb, S. J., & Johnson, M. H. (2020). Infant EEG theta modulation predicts childhood intelligence. Scientific Reports, 10(1), 11232. https://doi.org/10.1038/s41598-020-67687-y

Marriott Haresign, I., Phillips, E., Whitehorn, M., Noreika, V., Jones, E. J. H., Leong, V., & Wass, S. V. (2021). Automatic classification of ICA components from infant EEG using MARA. Developmental Cognitive Neuroscience, 52, 101024. https://doi.org/10.1016/j.dcn.2021.101024

Noreika, V., Georgieva, S., Wass, S., & Leong, V. (2020). 14 challenges and their solutions for conducting social neuroscience and longitudinal EEG research with infants. Infant Behavior and Development, 58, 101393. https://doi.org/10.1016/j.infbeh.2019.101393

Orekhova, E. (1999). Theta synchronization during sustained anticipatory attention in infants over the second half of the first year of life. International Journal of Psychophysiology, 32(2), 151–172. https://doi.org/10.1016/S0167-8760(99)00011-2

Perapoch Amadó, M., Greenwood, E., James, Labendzki, P., Haresign, I. M., Northrop, T., Phillips, E., Viswanathan, N., Whitehorn, M., Jones, E. J. H., & Wass, S. (2023). Naturalistic attention transitions from subcortical to cortical control during infancy. [Preprint]. Open Science Framework. https://doi.org/10.31219/osf.io/6z27a

Piazza, E. A., Hasenfratz, L., Hasson, U., & Lew-Williams, C. (2018). Infant and adult brains are coupled to the dynamics of natural communication [Preprint]. Neuroscience. https://doi.org/10.1101/359810

Räsänen, O., Kakouros, S., & Soderstrom, M. (2018). Is infant-directed speech interesting because it is surprising? – Linking properties of IDS to statistical learning and attention at the prosodic level. Cognition, 178, 193–206. https://doi.org/10.1016/j.cognition.2018.05.015

Richards, J. E. (2010). The development of attention to simple and complex visual stimuli in infants: Behavioral and psychophysiological measures. Developmental Review, 30(2), 203–219. https://doi.org/10.1016/j.dr.2010.03.005

Suarez-Rivera, C., Smith, L. B., & Yu, C. (2019). Multimodal parent behaviors within joint attention support sustained attention in infants. Developmental Psychology, 55(1), 96–109. https://doi.org/10.1037/dev0000628

Wass, S. V., Noreika, V., Georgieva, S., Clackson, K., Brightman, L., Nutbrown, R., Covarrubias, L. S., & Leong, V. (2018). Parental neural responsivity to infants’ visual attention: How mature brains influence immature brains during social interaction. PLOS Biology, 16(12), e2006328. https://doi.org/10.1371/journal.pbio.2006328

Xie, W., Mallin, B. M., & Richards, J. E. (2018). Development of infant sustained attention and its relation to EEG oscillations: An EEG and cortical source analysis study. Developmental Science, 21(3), e12562. https://doi.org/10.1111/desc.12562

Yu, C., & Smith, L. B. (2013). Joint Attention without Gaze Following: Human Infants and Their Parents Coordinate Visual Attention to Objects through Eye-Hand Coordination. PLoS ONE, 8(11), e79659. https://doi.org/10.1371/journal.pone.0079659

Yu, C., & Smith, L. B. (2016). The Social Origins of Sustained Attention in One-Year-Old Human Infants. Current Biology, 26(9), 1235–1240. https://doi.org/10.1016/j.cub.2016.03.026

Yu, C., Zhang, Y., Slone, L. K., & Smith, L. B. (2021). The infant’s view redefines the problem of referential uncertainty in early word learning. Proceedings of the National Academy of Sciences, 118(52), e2107019118. https://doi.org/10.1073/pnas.2107019118